# Exploring the ovine sperm transcriptome by RNAseq techniques. I Effect of seasonal conditions on transcripts abundance

**Irene Ureña[1], Carmen González[1], Manuel Ramón[2], Marta Gòdia[3], Alex Clop[3], Jorge H. Calvo[4], Mª Jesús Carabaño[1], Magdalena Serrano[1]***

**1** Departamento de Mejora Genética Animal, CSIC-INIA, Madrid, Spain, **2** IRIAF-CERSYRA, Valdepeñas, Ciudad Real, Spain, **3** Animal Genomics Group, Centre for Research in Agricultural Genomics (CRAG), CSIC-IRTA-UAB-UB, Catalonia, Spain, **4** Unidad de Tecnología en Producción Animal, CITA, Zaragoza, Spain

* malena@inia.csic.es

**Data Availability Statement:** The datasets generated and analysed in the current study are available at NCBI's BioProject PRJNA733107.

## Abstract

Understanding the cell molecular changes occurring as a results of climatic circumstances is crucial in the current days in which climate change and global warming are one of the most serious challenges that living organisms have to face. Sperm are one of the mammals' cells most sensitive to heat, therefore evaluating the impact of seasonal changes in terms of its transcriptional activity can contribute to elucidate how these cells cope with heat stress events. We sequenced the total sperm RNA from 64 ejaculates, 28 collected in summer and 36 collected in autumn, from 40 Manchega rams. A highly rich transcriptome (11,896 different transcripts) with 90 protein coding genes that exceed an average number of 5000 counts were found. Comparing transcriptome in the summer and autumn ejaculates, 236 significant differential abundance genes were assessed, most of them (228) downregulated. The main functions that these genes are related to sexual reproduction and negative regulation of protein metabolic processes and kinase activity. Sperm response to heat stress supposes a drastic decrease of the transcriptional activity, and the upregulation of only a few genes related with the basic functions to maintain the organisms' homeostasis and surviving. Rams' spermatozoids carry remnant mRNAs which are retrospectively indicators of events occurring along the spermatogenesis process, including abiotic factors such as environmental temperature.

## Introduction

It is now generally acknowledged that global warming is increasing and ecosystems, animal species diversity, and food security are at risk. These effects are especially noticeable in geographic areas with adverse environmental conditions, but also for nations in temperate zones where high-ambient temperatures are becoming an issue. The climate change negatively impacts the welfare and sustainability of livestock production and, consequently, food security in a rapidly growing population [1].

**Funding:** This work has been supported by the RTA2013-00041 INIA project.

In higher organisms almost all tissues, cell types, metabolic pathways and biochemical reactions are affected to some extent by heat stress and this is particularly true for the mammalian male germ cells, which are particularly thermo-sensitive. The sensitivity of these cells to environmental heat has been extensively studied. Likewise, heat stress has been linked to alterations in DNA, RNA abundance, protein synthesis, and chromatin packing in mice [2], boars [3] and rams [4, 5]. In most mammals, spermatogenesis occurs in the testicle, which is contained in the scrotum, and is exposed to a temperature that is approximately 3˚C lower than inside body temperature. Compared to other biological processes which typically occur at body temperature (~ 37˚C), spermatogenesis completely stops at this temperature [6].

Heat stress impacts on sperm quality and fertility in several ways. The exposure to high temperature and humidity may led to a reduction of the number of spermatozoa, sperm DNA damage and to a functional impairment [2], which will be accompanied by a transient period of partial or complete infertility. Thus, the climate can have a strong effect on reproduction efficiency, with obvious consequences on the animal's fitness [7] and the sustainability of the breeding sector [8].

Sperm cells are transcriptionally inert, and their RNAs were assumed to be afunctional debris that reflect the events that occurred during spermatogenesis [9]. Nevertheless, recent research have shown that sperm RNAs are related to germ cell development, sperm function, fertilization, early embryo development and even the offspring's phenotype [9, 10]. Moreover, sperm RNAs hold promising potential as biomarkers of sperm quality [11–13] and fertility [10, 14, 15]. The sperm contains a wide repertoire of coding and non-coding RNAs, including long as well as short non-coding RNAs [13, 16] which may have an important role in the sperm's fitness.

The aim of this study was to evaluate, for the first time, the effect of high environmental temperatures on the ovine sperm transcriptome by an RNA-seq approach. Paired sperm samples from 40 rams were collected under both heat stress (July, absolute maximum temperature 37.4˚C) and comfort temperatures (October, absolute maximum temperature 24.7˚C). A new bioinformatics pipeline, based on a cutting-edge software for single-cell RNA-seq analysis, was proposed to take into account for the particular features of sperm RNA, namely the very low abundances in most of the transcripts and a probably inflated large proportion of zero counts. Differential mRNA abundance between sperm samples collected under both climatic conditions were assessed, and genes and pathways involved, analyzed.

## Materials and methods

This study was carried under a Project License from the INIA Scientific Ethic Committee. Animal manipulations were performed according to the Spanish Policy for Animal Protection RD 53/2013, which meets the European Union Directive 86/609 on the protection of animals used in experimentation. We hereby confirm that the INIA Scientific Ethic Committee, which is named IACUC for the INIA, specifically approved this study.

Animals belonging to the CERSYRA-Valdepeñas (Spain) artificial insemination center, were raised in small groups in different barns and fed according to their requirements.

### Selection of rams, semen collection and evaluation of sperm quality traits

A total of 40 adult rams from Manchega sheep breed ageing from 16 to 90 months (average 34.3 ± SD 21.5) and belonging to CERSYRA artificial insemination center (Valdepeñas, Ciudad Real, coordinates: 38˚46'19.8"N 3˚23'20.1"W; 705 meters above sea level), were selected to conduct the experiment. Specialized professionals obtained two fresh ejaculates per ram during July 2016 and October 2016 to compare the effect of heat stress (HS) and comfort (C)

**Table 1. Weather conditions of the collection days obtained from the nearest meteorological station to the artificial insemination center were rams were located.**

| Date | Tm (˚C) | TMA (˚C) | tma (˚C) | Hr (%) |
|------|---------|----------|----------|--------|
| 25/07/2016 | 27.1 | 37.4 | 15.4 | 38.3 |
| 17/10/2016 | 15.7 | 24.7 | 7.7 | 68.7 |

Manzanares (Ciudad Real) meteorological station, (38˚59′46″N 3˚22′21″O; 661 meters above sea level. Tm = average temperature; TMA = absolute maximum temperature; tma = absolute minimum temperature; Hr = average relative humidity.

environmental conditions on the sperm's transcriptome. Table 1 includes the climatic parameters existing at the collection dates taken from Manzanares (Ciudad Real) meteorological station (coordinates: 38˚59′46″N 3˚22′21″O; 661 meters above sea level). Rams were handled in conditioned boxes inside the barn, with natural ventilation and similar feeding and management. The animals are allowed to go outside the stables daily, but avoiding the hottest hours of the summer months. The climatic data recorded (Table 1) reflects the outdoor conditions, in order to display the differences between the hot and comfort seasons in temperature and relative humidity.

Ejaculates were collected with an artificial vagina. This device is connected to a graduate crystal falcon to measure the total ejaculate volume and to keep the ejaculate immersed in a 37˚C water bath. Ejaculates were maintained at 37˚C for a maximum of 1h until spermatozoa purification.

## Sperm quality traits assessment

Routine sperm quality traits were measured on all ejaculates used in this work, i.e. on 80 ejaculates from 40 rams collected in July (heat stress) and October (comfort). Three traits were assessed for each ejaculate: ejaculate volume (mL) which was measured using a graduated collection tube, ejaculate concentration (spermatozoa x 106/mL) which was determined using a standard spectrophotometer (Thermo Electron coorp. Heλios), and mass motility, which was assessed for undiluted semen under a microscope. Mass motility was scored subjectively, based on wave motion, on a continuous scale from 0 (no motion) to 5 (frequent rapid and vigorous waves). A fourth derived trait, the number of spermatozoa (SPZ) was computed as the product of the ejaculate volume and sperm concentration. Table 2 shows the values obtained for these traits in each ejaculate collection (heat stress and comfort). A T-test was performed to determine if significant differences exists between sperm traits collected in the heat and comfort seasons.

## Sperm purification

The purification of the spermatozoa cells was performed using 5ml of BoviPureTM (Nidacon International, Mölndal, Sweden) diluted to a final ratio of 65% (v/v) with BoviDiluteTM

**Table 2. Mean and standard deviation of sperm traits routinely assessed at the AI center between seasons.**

| Season | Sperm Mass Motility (0–5) | Ejaculate Volume (ml) | Sperm concentration (spz ×10$^6$/ml) | Number of spermatozoa |
|--------|---------------------------|------------------------|--------------------------------------|------------------------|
| Comfort (17/10/2016) | >4 | 1.06 ± 0.37 | 4071.4 ± 789.8[a] | 4523 ± 2327.0 |
| Heat stress (25/07/2016) | >4 | 1.17 ± 0.41 | 3596.5 ± 840.7[b] | 4334 ± 2136.1 |

[a,b] different superscripts indicate significant differences between seasons (p<0.05).

(Nidacon International, Mölndal, Sweden) in 15 ml nuclease-free tubes. The volume of sperm that was layered on top of the layer Bovipure™ was decided in accordance with its concentration/volume ratio, with extended ejaculate with BoviWash™ (Nidacon International, Mölndal, Sweden) and, in all cases, with a concentration around of 233x106 cells/ml. Purification was performed by a 300xg centrifugation (without brake option) during 25 min at room temperature. After centrifugation, all the upper phases were aspirated with sterile Pasteur pipettes and the sperm pellet was transferred to a new 15 ml nuclease-free tube, washed with 10ml Bovi-Wash™ and 1500xg centrifuged (with brake option) for 10 min at room temperature. The supernatant was removed and the pellet was resuspended in 1 ml of BoviWash™. Resuspended pellets were transferred to 1.8 ml cryotubes and immediately frozen in liquid nitrogen. Samples were stored at -80˚C until further use for RNA extraction.

## RNA extraction, DNAse treatment and cDNA synthesis

Total RNA was extracted from 80 purified sperm samples belonging to 40 different rams (two sperm samples per ram). The starting material was 75µl of diluted semen with a concentration < 90 x 106 sperm cells/sample. RNA was isolated with TRIsure™ reagent (Bioline, Taunton, MA, USA) following specific protocol for cells in suspension of TRIsure instructions (TRIsureTM protocols guide). The RNA samples were resuspended in 60µl DEPC-water and were quantified with Nanodrop spectrophotometer (Thermo Scientific, Wilmington, DE, USA).

RNA samples were subjected to DNase treatment with RQ1 RNAse-Free DNase (Promega Corporation, Madison, WI, USA) following the manufacturer's instructions with some modifications: 2 units of DNase per 0.5–1µg of RNA sample and 1.1 µl of 10x Reaction Buffer at 37˚C for 30 min were incubated. Finally, to inactive the DNase 2µl of DNase Stop Solution was incubated at 65˚C for 10 min. To analyze overall RNA fragmentation and the absence of intact ribosomal 18S and 28S RNAs in sperm, RNA samples were evaluated on a 2100 Bioanalyzer using Agilent RNA 6000 Pico kit (Agilent Technologies, Santa Clara, CA, USA).

Reverse transcription was performed from 500ng total sperm RNA with 1µl random hexamer primers and 1µl Oligo dT using the Improm-II Reverse Transcription System (Promega Corporation, Madison, WI, USA) following the manufacturer's instructions. All cDNA samples concentrations were normalized to 50ng/µl.

## PCR and qPCR controls

Some PCR controls were performed to asses that RNA was devoid of genomic DNA contamination and somatic cells specific genes.

- Immediately after sperm RNA extraction and to determine if sperm RNA were contaminated with somatic cells DNA, PCRs with ovine-specific designed primers (exon-exon) of the CD4 gene (Tcell surface glycoprotein CD4) were performed (Table 3).

- After DNase treatment of sperm RNA, PCRs using specific primers bovine and ovine (exon-exon) of the PRM1 gene (Protamine 1) based in those designed by Feugang and colleagues [14] (Table 3) were performed to check genomic DNA contamination.

- Finally, and after the reverse transcription step, PCRs with ovine-specific designed primers (intra-exonic) of the PTPRC gene (Protein Tyrosine Phosphatase Receptor type C) (Table 3) were used to check somatic cell RNA contamination.

To further verify with greater sensitivity that cDNA was derived only from the mature sperm RNA battery and not from somatic cell contamination, two absolute qPCR assays were

**Table 3. Ovine-specific primer sequences for PCR and qPCR amplifications of *CD4*, *PTPRC* and *PRM1* genes, PCR melting temperatures and product size.**

| Gene | Primer sequence | Tm(˚C) | Product size |
|------|-----------------|--------|--------------|
| *CD4* | **F** 5'-CTC TCT TAG GCA CCT GTT CTT G-3' | 58˚C | gDNA 196 bp |
|  | **R** 5'-CAC CAC TGC TTT TCC CTG A-3' |  | RNA/cDNA 73bp |
| *PTPRC* | **F** 5'-CGC CCA GAA TGG ACA AGT AA-3' | 58˚C | 157bp |
|  | **R** 5'CTT GGT GCC TCC AGC CTC-3' |  |  |
| *PRM1* | **F** 5'-AGA TGT CGC AGA CGA AGG AG-3' | 60˚C | gDNA 220 bp |
|  | **R** 5'-AGT GCG GTG GTC TTG CTA CT-3' |  | RNA/cDNA 117bp |

F: forward; R: reverse; Tm melting temperature.

carried out targeting PRM1 and PTPRC genes (Table 3). The standard curves were performed based on 5 serial dilutions (a departure concentration of 50 ng/µl) of a sperm cDNA stock (mixture of all samples for the two time points) for PRM1 and PTPRC genes to check primer specificity and to confirm the presence of a unique PCR product. Reactions were performed in a total volume of 25µl, 0.3µM of each primer and 10 µl of SYBR Green I Master kit (Roche, Switzerland) in triplicate on a LightCycler® 480 (Roche, Switzerland) following manufacturer's cycling parameters.

To perform qPCR of the 80 samples (40 samples on July and 40 on October) we started from 100ng (2µl) of sperm cDNA and blood cDNA (positive control) and one negative control, all of them in triplicate. The thermal profile was: pre-incubation 95˚C for 10 min, amplification 40 cycles of 95˚C 15 sec, 60˚C (PRM1 gene) or 58˚C (PTPRC gene) 30 sec and quantification 72˚C 1 sec (quantification single point type). The corresponding mRNA levels were measured and analyzed by the average of the Cp (Crossing point) raw data.

## Library construction, RNA sequencing and RNA-seq mapping analysis

A total of 64 samples were finally used for library prep, this included 24 rams with ejaculates both in HS and C, and 4 and 12 rams with ejaculates collected only in HS and C, respectively. Sperm RNA-seq libraries were constructed with the SMARTer® Stranded Total RNA-Seq Kit —Pico Input Mammalian (Takara Bio USA, Inc.) according to the manufacturer's instructions, which is compatible with pico-gram inputs of total RNA from high quality or partially degraded samples (250 pg –10 ng). Sequencing of stranded mRNA was performed in a High-Seq2000 Illumina platform to obtain 75bp long paired-end reads.

Quality control of the raw and trimmed paired-end reads were performed with FastQC v.0.11.7 (https://www.bioinformatics.babraham.ac.uk/projects/fastqc/). Trimming was performed with Trimmomatic v.0.38 [17] to remove low quality reads and adaptors. Trimmed reads were mapped to the sheep reference genome (Oar_v3.1, assembly GCA_000298735.1) with HISAT2 v.2.0 [18] with default parameters except "–max seed 30" and "–k 2". Picard MarkDuplicates was used to check the percentage of duplicate reads (http://broadinstitute.github.io/picard/). Mapped transcripts were assembled individually with Stringtie v.1.3.3b [19]. Annotation (Ensembl Release 104) was performed by using the -G option. The output included reference transcripts as well as any novel transcripts that were assembled.

## Statistical analysis

To conduct the differential abundance (DA) analysis, a pre-filtering step was performed following the DESeq2 manual [20] such that there must be at least five samples with normalized counts greater or equal to 5 to keep the gene in. The transcriptome profiles showed very low

RNA abundances in most of the transcripts and a probably inflated large proportion of 0 counts. This situation is somewhat similar to the scenario that occurs in single cell RNAseq studies. For this reason, we applied the methodology proposed by Van den Berge et al. [21] using the ZINB-WaVE 1.4.0 software, to extract low-dimensional signal from noisy, zero inflated data (from dropouts and bursting) [22] in conjunction with DESEq2 v.1.20.0 [20]. The data was analyzed using a zero-inflated model and performing inferences on the count component of the model, which is equivalent to standard negative binomial regression where excess zeros are down weighted based on posterior probabilities (weights) inferred from the ZINB-WAVE method. ZINB-WaVE model included only the intercept (V = 1) and a population covariate (October or July) in X. Epsilon was set to 1010 and the observational weights were computed with the number of unknown covariates K = 0, i.e., no latent variables were inferred. Batch covariate "run" was not included since the batch effect does not seem to be associated with the sequencing depth. In addition, when the run effect was included as a sample-level covariate in X, ZINB-WaVE yielded similar results (see S1 Fig).

DA genes between the two environmental periods (HS and C) were investigated. DESeq2 software was employed following [21] parameters, using the count data matrix obtained from Stringtie, the phyloseq normalization procedure and the likelihood ratio test implemented in nbinom LRT. The effect of the animal was also included in the DESEq2 model using a multi-factor design which includes sample information. The minimum expected count was set to 10–6 following [21]. Genes were considered to be DA when the significance level (adjusted p-value for multiple comparisons) was lower than 0.05, after the Benjamini-Hochberg correction (padj) and the absolute value of the Log2FoldChange was higher or equal to 1.5. Annotation of the DA genes between the two climatic conditions was retrieved with BiomaRt (https://www.ensembl.org/biomart).

Gene set enrichment analysis of the genes detected was performed with Gene Set Enrichment Analysis (GSEA) (https://www.gsea-msigdb.organd) [23] through the OmicsBox platform (https://www.biobam.com/omicsbox). In addition, Ingenuity Pathway Analysis (IPA) (Ingenuity Systems, Qiagen, California) was used in order to identify and characterize canonical pathways, biological functions, gene networks and regulators for genes detected in sperm samples.

## Results

### RNA extraction, sequencing output and genes detected in sperm samples

We obtained sufficient RNA yield from 64 ejaculates, 28 collected in July (HS) and 36 obtained in October (C). Both conditions were available in ejaculates from 24 rams. RNA extraction yielded an average of 2μg per ejaculate and the concentration was 50–200 ng/μl (measured with Nanodrop). These RNAs were devoid of intact ribosomal 18S and 28S RNA with RIN values below 2.6 and were free of gDNA and RNA from somatic cell sources based on PCR and qPCR amplification of *CD4*, *PRM1* and *PTPRC*, as described above. No PCR amplifications of genomic *CD4*, genomic *PRM1* and *PTPRC* RNA were obtained. As expected, qPCR results showed the amplification of the *PRM1* gene (Cp ≅25) and no signal for the *PTPRC* gene (Cp>37). Blood cDNA from these same rams, included to monitor expression of *PTPRC*, showed a clear amplification of this gene (Cp ≅25).

High throughput sequencing yielded an average of 29.4 million of read pairs per sample (S1 Table, shows the sequencing results), from which, an average of 24.3 million (83% of the sequences) were kept after trimming and 16.7 million were successfully aligned to the sheep genome. Samples with less than 60% of mapped reads were excluded for further analysis. In total 61 RNA samples were kept, 27 for HS and 34 for C. Twenty four animals had samples in both conditions in the final dataset.

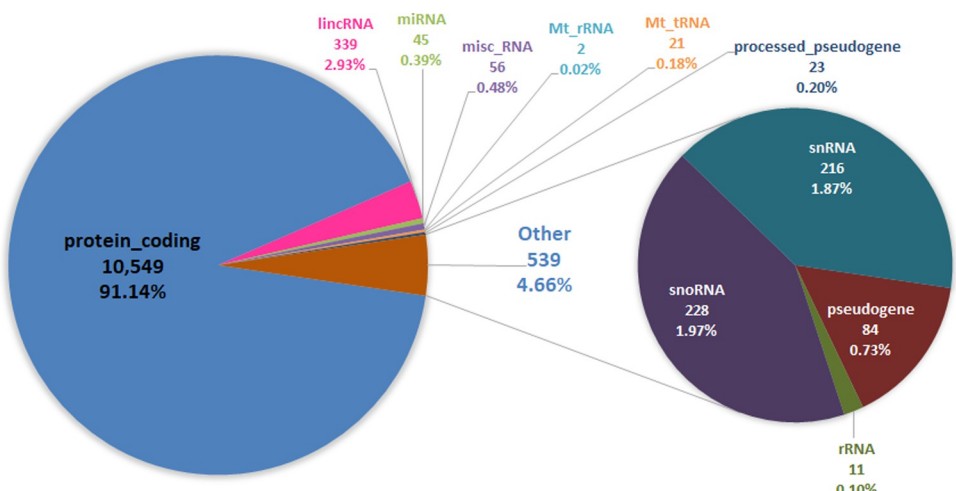

**Fig 1. Types of genes and transcripts detected in rams' sperm samples.** Number and percentage of each type of gene are included. Mt-tRNA: mitochondrial transfer RNA; rRNA: ribosomal RNA; snRNA: small nuclear RNA; snoRNA: small nucleolar RNA; miRNA: microRNA precursors; misc_RNA: miscellaneous other RNA; lincRNA: Long intergenic non-coding RNAs.

A total of 11,895 transcripts and 11,574 genes were identified in the ram's sperm transcriptome (S2 Table). Most transcripts and genes were protein coding, 10,856 and 10,549, respectively, but other RNA types were also found (Fig 1). The top 20 most abundant protein coding genes with average between 479 and 1,968 CPM (count per million) are shown in Table 4. This list included, genes related with spermatid maturation, flagelled sperm motility and acrosome reaction (*SPEM1* and *SPEM2*); one testis specific kinase (*TSSK6*); 2 A-kinase anchoring proteins (*AKAP12* and *AKAP1*), 2 kinesins involved in testis transcription signaling (*KIF21B* and *KIF17*), 1 gene which controls the metabolism of fatty acids at different levels (*LPIN1* lipin1) and a gene encoding a major component of sperm tail outer dense fibers (*ODF2* outer dense fiber of sperm tails 2).

## Gene ontology analysis and canonical pathways of the ram's sperm transcriptome

Gene set enrichment analysis by GSEA gene of the 11,574 genes detected in the sperm samples, showed 18 GO terms (p-value < 0.05 and FDR q-val < 0.1). (S3 and S4 Tables). Fig 2 shows a bar chart with GSEA gene ontology analysis for all the genes detected with an FDR ≤ 0.10.

Pathway enrichment analysis of the 11,574 sperm genes annotated in the sheep genome showed that the most overrepresented canonical pathways were related to protein kinase A signaling, Protein ubiquitination, Molecular mechanisms of cancer, RAR activation and AMPK signaling (S5 Table). Among the top canonical pathways highlighted by the IPA analysis, some were related with male reproductive functions (Germ Cell-Sertoli Cell Junction, Sperm Motility, Androgen Signaling), DNA damage response (Role of BRCA1 in DNA Damage Response, Cell Cycle G2/M DNA Damage Checkpoint Regulation, DNA damage-induced 14-3-3σ Signaling) and Oxidative stress response (NRF2-mediated Oxidative Stress Response).

## Differentially abundant genes related to heat stress

Differential abundance analysis between sperm samples collected under HS vs C (S6 Table) yielded 236 differentially abundant genes (padj < 0.05) but only 98 with a |log$_2$FoldChange| ≥

**Table 4. Twenty most abundant protein coding genes detected in rams sperm samples.**

| Chrom | Ensembl ID | Average CPM | sd CPM | Gene ID | Gene description |
|---|---|---|---|---|---|
| 1 | ENSOARG00000017870 | 1,968 | 1,273 | HDLBP | high density lipoprotein binding protein |
| 11 | ENSOARG00000013274 | 1,410 | 754 | SPEM2 | SPEM family member 2 |
| 8 | ENSOARG00000003112 | 1,268 | 919 | AKAP12 | A-kinase anchoring protein 12 |
| 12 | ENSOARG00000017121 | 1,179 | 888 | KIF21B | kinesin family member 21B |
| 2 | ENSOARG00000008948 | 1,107 | 751 | KIF17 | kinesin family member 17 |
| 26 | ENSOARG00000004167 | 985 | 605 | NGLY1 | N-glycanase 1 |
| 3 | ENSOARG00000016144 | 903 | 572 | LPIN1 | lipin 1 |
| 4 | ENSOARG00000003244 | 857 | 548 | CCDC136 | coiled-coil domain containing 136 |
| 5 | ENSOARG00000007818 | 846 | 438 | TSSK6 | testis specific serine kinase 6 |
| 22 | ENSOARG00000004357 | 808 | 474 | NEURL1 | neuralized E3 ubiquitin protein ligase 1 |
| 11 | ENSOARG00000013266 | 753 | 396 | SPEM1 | spermatid maturation 1 |
| 1 | ENSOARG00000007404 | 713 | 464 | MROH7 | maestro heat like repeat family member 7 |
| 11 | ENSOARG00000007626 | 679 | 417 | AKAP1 | A-kinase anchoring protein 1 |
| 6 | ENSOARG00000014418 | 617 | 337 | GRK4 | G protein-coupled receptor kinase 4 |
| 3 | ENSOARG00000010125 | 611 | 365 | ODF2 | outer dense fiber of sperm tails 2 |
| 5 | ENSOARG00000013036 | 599 | 350 | CSNK1G2 | casein kinase 1 gamma 2 |
| 11 | ENSOARG00000012958 | 496 | 305 | ACE | angiotensin-converting enzyme |
| 1 | ENSOARG00000006234 | 496 | 341 | ISG20L2 | interferon stimulated exonuclease gene 20 like 2 |
| 9 | ENSOARG00000013994 | 485 | 261 | LYPLA1 | lysophospholipase 1 |
| 20 | ENSOARG00000014364 | 479 | 261 | PIM1 | Pim-1 proto-oncogene, serine/threonine kinase |

Chrom = chromosome; CPM = counts per million.

1.5. From these, 92 were downregulated and 6 upregulated under HS conditions. Fig 3 shows a scatter plot (DESeq2 plotMA) for sperm differential abundance genes between control (comfort) and heat stress climatic conditions.

The protein coding genes showing differential abundance between HS and C conditions (padj < 0.05) with a |log2FoldChange| ≥ 2.0 are shown in Fig 4. Only 6 genes showed an upregulated transcriptional activity induced by HS conditions, *FAM126B* (family with sequence

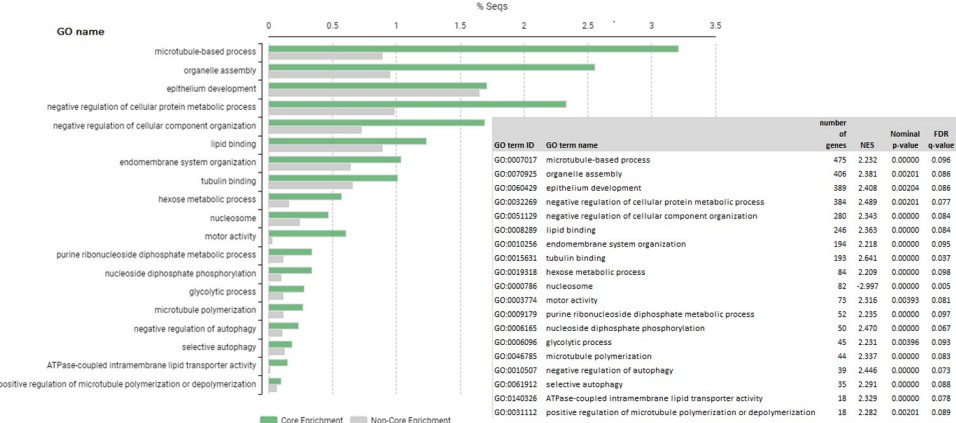

**Fig 2. Gene Set Enrichment Analysis (GSEA) bar chart for all the genes detected in sperm samples.** Top Go terms by the Normalized Enrichment Score (NES) for an FDR < 0.1.FDR = False discovery rate. % Seqs = percentage of sequences in each GO term.

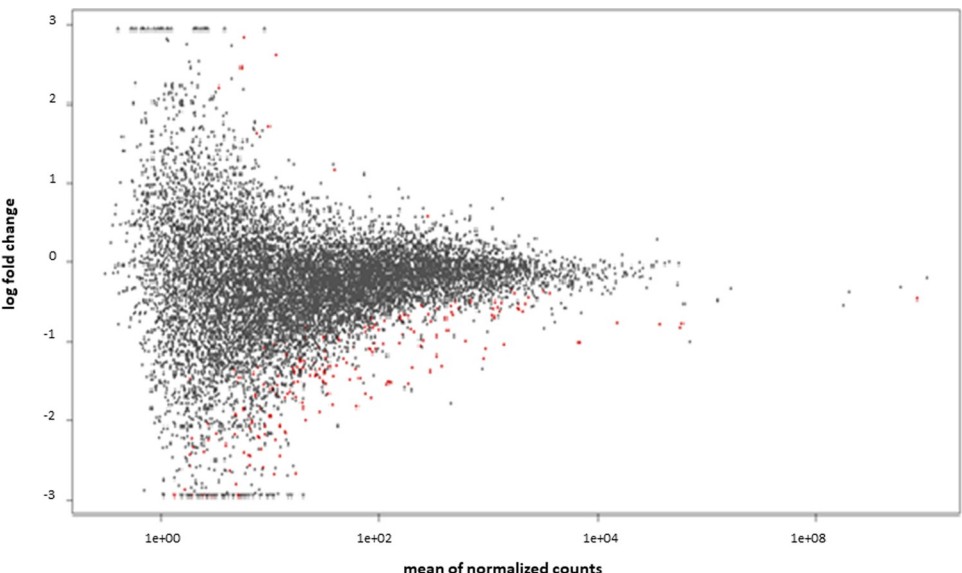

**Fig 3. Scatter plot (DESeq2 plotMA) for sperm differential abundance genes between control (October) and heat stress (July) climatic conditions.** Red dots show the significant differential abundance genes for an adjusted p-value <0.05.

similarity 126 member B), *PNLIPRP3* (Triacylglycerol lipase), *MS12* (Musashi RNA binding protein 2), *RASIP1* (ras interacting protein 1), *RGS2* (Regulator of G protein signaling 2) and *FAM90A1* (family with sequence similarity 90 member A1). The downregulated genes under high temperatures included *ALCAM* (Activated leukocyte cell adhesion molecule), *NOTCH2* (Notch receptor 2), *URI1* (URI1 prefolding like chaperone), *CCT4* (Chaperonin containing TCP1 subunit 4), *PRPF38B* (pre-mRNA processing factor 38B) and *SRSF10* (Serine and Arginine Rich Splicing Factor 10), among others.

## Gene ontology analysis, canonical pathways and gene networks of the differentially abundant genes between conditions

Gene set enrichment analysis of the 98 DA genes (log2FoldChange $\geq$ |1.5|, p-value < 0.05 and FDR q-value < 0.1) (Fig 5), yielded 5 biological process GO terms: GO:0019953 sexual reproduction, GO:0051704 multi-organism process, GO:0033673 negative regulation of kinase activity, GO: 0032269 negative regulation of cellular protein metabolic process and GO:0071900 regulation of protein serine/threonine kinase activity. Genes included in these GO terms were the upregulated *RGS2* (regulator of G protein signaling 2) and *RASIP1* (Ras interacting protein 1) and the downregulated *SHCBP1L* (SHC binding and spindle associated 1 like) and *CD46* (Membrane Cofactor Protein CD46, Trophoblast-Lymphocyte Cross-Reactive Antigen, CD46 molecule). S7 Table, shows the results of the gene set enrichment analysis of the 98 HS DA genes.

A summary of the Ingenuity pathway analysis of the 98 DA genes is shown in S7 Table. The main significant pathways found for DA genes were Oxidative Phosphorylation (inhibited) and Mitochondrial Dysfunction which involved both mitochondrial molecules such as ATP5F1B, MT-ATP6, MT-ND2 and MT-ND4L. Cardiac Hypertrophy Signaling was also identified as inhibited and includes DIAPH2, DLG1, IL17RC, PDE1C and TGFB3 molecules.

A total of 10 networks were identified by IPA for the DA genes. The top 5 networks with score > 15 are shown in Table 5. Network 3 was related to Reproductive System Development

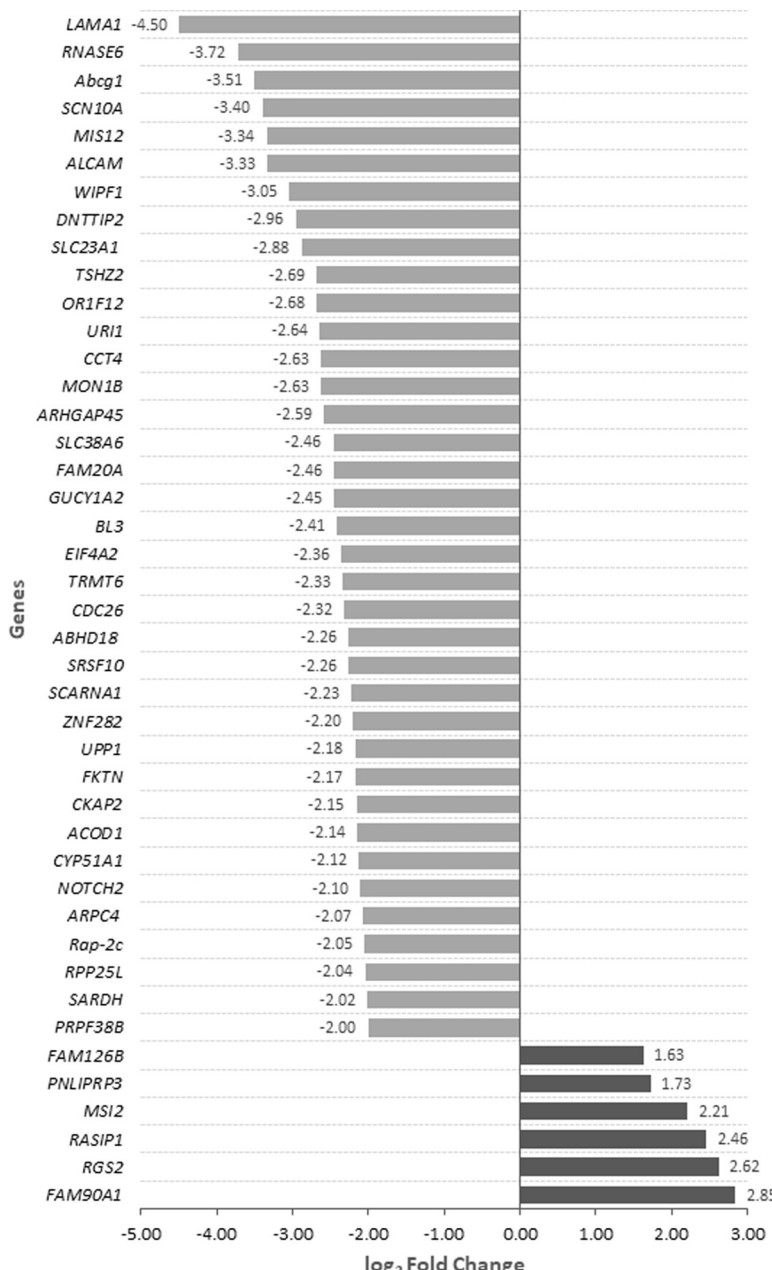

**Fig 4. Upregulated (dark grey) and down-regulated (light grey) protein coding genes under heat stress conditions.** The DA genes passed the filters of padj < 0.05 and absolute log2FoldChange ≥ 1.5. In this figure, we only show genes that presented padj < 0.05 and absolute log2FoldChange ≥ 2.

and Function, and involves 12 focus molecules. Top diseases and disorders list identifies reproductive system disease in the third place involving 61 molecules. Reproductive system development and function was in the top of the physiological system development and function list including 8 molecules. Activated molecules identified were RGS2, RASIP1, PNLIPRP3, FAM126B, ABCA5 and MPRIP. Inhibited molecules group included SCN10A, ALCAM, WIPF1, DNTTIP2, TSHZ2, URI1, MON1B, ARHGAP45, SLC38A6 and FAM20A. The most relevant upstream regulators found for the DA genes include enzymes, SIRT3 (activated), and

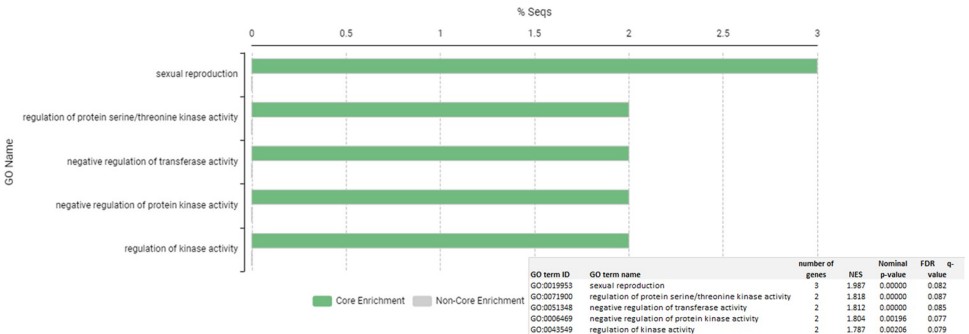

**Fig 5. Gene Set Enrichment Analysis (GSEA) bar chart for the 98 differential abundant genes between climatic conditions.** Top Go terms by the Normalized Enrichment Score (NES) for an FDR < 0.1. FDR = False discovery rate. % Seqs = percentage of sequences in each GO term.

transcriptional regulators, STAT3 (inhibited) and TCF4 (inhibited). Some of the regulator effects are related to DNA repair, spermatogenesis, cell survival, embryonic cell death, protein metabolism and transcription.

## Discussion

Historically, spermatozoa have been considered transcriptionally silent and a mere carrier of the paternal genome to the oocyte. However, today we know that sperm carries a large and varied collection of RNAs, including messenger RNA (mRNA), micro-RNA (miRNA), interference RNA (iRNA), and antisense RNA [9], associated with functions related to spermatogenesis [10], sperm quality traits [11, 24] and early embryonic development [10, 25]. Two facts

**Table 5. Top five networks enriched with differential abundant (DA) genes.**

| Network ID | Molecules in Network | Score | Focus Molecules | Top Functions |
|---|---|---|---|---|
| 1 | AGPS, AKIRIN2, Akt, ALCAM, AMOT, ARID3A, ATP5F1B, BANK1, CD6, Creb, CYP51A1, DLG1, ERK1/2, estrogen receptor, HDL-cholesterol, ING4, KLHL24, NCOA3, NFkB (complex), NOTCH2, OTUB2, Plc beta, PTPRZ1, RALGDS, RASGRF2, RGS2, RNA polymerase II, SCAF8, SLC25A3, Sos, TCR, TGFB3, TSPAN33, TTC39B, URI1 | 34 | 18 | Hematological System Development and Function, Humoral Immune Response, Lymphoid Tissue Structure and Development |
| 2 | AGBL2, APEX1, BUD23, CDK7, CEBPD, CKAP2, CREBBP, CTSD, CYP51A1, DNTTIP2, DTX1, EXT2, FZR1, GREB1, IL17RC, KDM4C, LDHA, LRPPRC, LSS, mir-17, MTDH, NAP1L4, PDGFA, POU5F1, PPARA, Rnr, SARDH, SQLE, STPG4, TOP2A, TP53, TP73, TRMT6, TSHZ2, ZNF282 | 20 | 12 | Cancer, Neurological Disease, Organismal Injury and Abnormalities |
| 3 | **ACOD1, ADAM2, Adam3, ALK, ALKBH1, ATXN7, CD19, CD9, DAP3, FER, KAT2A, KDM3A, KIF5B, MBIP, MT-ATP6, mt-Atp8, MT-ND2, MT-ND4L, MYC, NSUN3, PSEN1, SCN10A, SGF29, STAT3, SUPT3H, TADA2A, TAF1A, TAF1B, TAF1C, TAF9, TMF1, TXNDC11, WIPF1, XBP1, YEATS2** | 20 | 12 | Cell-To-Cell Signaling and Interaction, Cellular Assembly and Organization, **Reproductive System Development and Function** |
| 4 | AQP1, AQP9, CRPPA, DAG1, DNAH7, EIF1, FAM20A, FGF1, FKTN, GPR158, IFNG, Interferon alpha, KPNA4, LAMP3, LARGE1, MAML1, MAPK1, MBTD1, MDK, NFKBIA, NUP50, NUPR1, POMGNT1, PRICKLE1, RELA, RNF103, SPATS2L, SPRTN, SREBF1, SUPT20H, SUPT3H, SYNE2, VCAN, ZC3HAV1, ZMYND11 | 20 | 12 | Developmental Disorder, Hereditary Disorder, Metabolic Disease |
| 5 | Alpha catenin, AQP1, CAPZB, CDC73, Cdk, CRKL, CTNNB1, CTNND1, CXCL8, DIAPH2, DOCK7, ERG, ETV3, FCER2, GBF1, HDAC2, IFRD1, IGF1,IKZF1, ITGB1, MAML1, mir-192, MMP14, MPRIP, MTPN, MYL12B, PDE1C, Pdgfr, PRIM2, PSMB8, RAB11FIP2, RAB1B, RASIP1, RHOA, TMED1 | 16 | 10 | Cell-To-Cell Signaling and Interaction, **Cellular Movement,** Connective Tissue Development and Function |

evidenced that sperm RNAs are not merely residual products from the spermatogenesis process. First, there are evidences of translational activity in the sperm cells [26]. Second, sperm mRNAs have been associated with semen quality traits as sperm count and motility and pregnancy outcomes [27], whereby its study might help in the diagnosis of male infertility [10]. These facts imply that the presence of sperm RNA has fitness consequences for both males and females, and it is there because of its adaptive value [28].

## Ovine sperm transcriptome

In this work, by combining the use of SMARTer™ Ultra Low RNA libraries, the HighSeq2000 Illumina RNA sequencing and the ZINB-based Wanted Variation Extraction (ZINB-WaVE) method to successfully characterize the transcriptome of Manchega rams sperm cells, we have identified 11,895 transcripts from 11,574 genes annotated in the sheep reference genome. ZINB-WaVE method of Risso and colleagues [22], which is a general and flexible framework for the extraction of a low-dimensional signal from scRNA-seq read counts accounting for zero inflation (i.e., dropouts and bursting), efficiently identifies excess zeros and provides gene-specific weights to unlock bulk RNA-seq pipelines (DESeq2) for zero-inflated data.

Most transcripts belong to protein coding genes (10,856), but also mitochondrial RNAs, miRNAs, lincRNAs, snRNAs and snoRNAs were found. While the functions of sperm miRNAs and ncRNAs in mammalian biology are yet to be determined, data from mouse and humans suggests that these RNAs regulate gene expression in the early zygote either by direct interaction with mRNA or via epigenetic mechanisms [29, 30] and may also influence semen fertility [16, 31]. From all transcripts (11,895), only 43 displayed an average abundance above 400 CPM and were thus considered as representative of the highly abundant sperm transcript class (S2 Table).

The top 20 most abundant protein coding genes (Table 4) were involved in different biological processes, predominantly those specific of sperm cells. The most abundant was the *HDLBP* (High Density Lipoprotein Binding Protein) gene, also named Vigilin. This gene has been related with the larval development in *Caenorhabditis elegans* by interacting with miRNAs to regulate gene expression dynamics for development [32], but also with DNA damage repair, chromatin condensation and gene silencing [33]. In humans, the highest expression levels of Vigilin gene was found in testis (284 TPM), prostate (208 TPM), placenta (202 TPM) and ovary (202 TPM) [34], therefore this gene may be essential for reproductive biological processes. Among these 20 top most abundant genes we found *SPEM1*, *SPEM2*, *TSSK6*, *AKAP12*, *KIF21B*, *KIF17* and *ODF2*. *SPEM1* (Spermatid Maturation 1) and *SPEM2* (SPEM Family Member 2) genes are related to spermatid maturation and flagelled sperm motility, and both are linked to reproductive disorders such as male infertility, spermatogenic failure and oligospermia [35]. *TSSK6* (Testis specific serine kinase 6, also named SSTK) gene is a testis specific kinase which is essential for male fertility [36] and specifically interacts with HSP90-1beta, HSC70 and HSP70 proteins to achieve its kinase activity [37]. Targeted deletion of the *SSTK* gene from the mouse genome resulted in male sterility due to impaired postmeiotic chromatin condensation and abnormal spermiogenesis [36]. Also high levels of *TSSK6* RNA were found in cattle sperm cells [38]. *AKAP12* (A-kinase anchoring protein 12) and *AKAP1* (A-Kinase Anchoring Protein 1) were also found highly expressed in the ram's sperm. In human and mouse, *AKAP12* has its highest expression levels in testis [39] and has been related in mice with both, male and female fertility [40]. This gene has an essential role in numerous aspects of cell biology including cell adhesion, cell morphology, cytokinesis, and migration [41]. *AKAP12* is involved in blood coagulation and protein kinase A signaling regulation biological processes. Both genes appear to be key molecules in the biochemical machinery regulating the

sperm motility [42]. Two kinesins, *KIF21B* and *KIF17*, showed high abundance in the ram's sperm samples. In boars' sperm, also high abundance of *KIF17* transcripts was found [3]. During spermatogenesis, male germ cells undergo massive differentiation and rapid polarization that require transport and specific localization of vesicles, proteins, mRNAs, and organelles. A number of kinesin and dynein superfamily motor proteins have been proposed to carry out transport processes in the testis. *ODF2* (Outer dense fiber of sperm tails 2) gene, seems to be a major component of sperm tail outer dense fibers (ODF), which are cytoskeletal structures that surround the axoneme in the middle piece and principal piece of the sperm tail and modulates sperm motility. Odf2 protein play a key role in the formation of motile sperm flagella [43].

GSEA and IPA analyses using as input all the genes found in the ram's sperm transcriptome, identified a high enrichment of genes and GO terms related with the microtubule-based processes. Sperm with normal morphology and motility is essential for normal male fertility. Microtubules and microfilaments play pivotal roles in the maintenance of these sperm qualities during different phases of sperm production and maturation [44]. In spermatogenic cells, microtubules are necessary for several processes, including the assembly of flagella in spermatids and the generation and maintenance of motility in mature spermatozoa [44]. Microtubules, which are composed of α and β tubulins, are the central element of cilia and flagella, and therefore any defects in their structure can lead to infertility. A number of genes associated with microtubular formation and function were found in the ovine sperm transcriptome herein assessed. The *HOOK1* (Hook Microtubule Tethering Protein 1) gene encodes a protein that binds to microtubules and cargoes. This protein belongs to the family of Hook proteins, consisting of Hook1, Hook2 and Hook3 in mammals. We found *HOOK1*, *HOOK2* and *HOOK3* genes in the core enrichment of the GO_0007017 term. *HOOK2* plays a role in primary cilia morphogenesis while *HOOK3* has been shown to participate in the localization of the Golgi complex. *HOOK1* is suggested as a candidate for decapitation defects and teratozoospermia [45]. Mutations of several genes involved in microtubule biological processes and molecular functions have been associated with male infertility by disrupting microtubular functions during the formation of sperm affecting normal sperm motility [45]. The *KATNB1* (Katanin Regulatory Subunit B1) gene encodes the protein p80, and a missense mutation of this gene causes male infertility in mice characterized by oligoasthenoteratozoospermia and virtual absence of progressive motility [46]. The *SEPT12* (Septin12), *KIF3A* (Kinesin family member 3A), *RSPH9* (Radial spoke head component 9), *DNAL1* (Dynein axonemal light chain 1), *DNAJB13* (DnaJ heat shock protein family Hsp40 member B13) and *ZMYND10* (Zinc finger MYND-type containing 10) genes, among others, are involved in sperm flagellar motility and the maintenance of the structural integrity of human mature sperm. Deficiencies of these proteins cause marked motility defects and morphologically abnormal and disorganized flagellar structures [47].

Among the enrichment ontology functions, we also found a good number of biological processes related with "autophagy" (S3 and S4 Tables). Autophagy is an evolutionarily conserved process by which cellular components can be transported into lysosomes for degradation and recycling thereby facilitating cellular homeostasis and survival [48]. Its activity is fundamental to many processes across the reproduction spectrum from development of the primordial follicle and spermatozoa to embryogenesis, placental development and maintaining uterine quiescence during pregnancy [49]. Dysregulation of autophagy has been linked to mice male infertility; testosterone synthesis in Leydig cells [50], ectoplasmic specialization of Sertoli cells [51], acrosome biogenesis [52] and round spermatid specific chromatoid body integrity [53]. Autophagy processes included the activity of the ATG7 (autophagy related 7) gene, which has been associated to acrosome formation, flagella biogenesis and cytoplasm removal [54].

The organelle assembly GO term involves the aggregation, arrangement and bonding of a set of components to form the nucleus, mitochondria, plastids, vacuoles, vesicles, ribosomes and the cytoskeleton, but not the plasma membrane. Child terms related with reproductive functions are the male and female pronucleus assembly and the acrosome and flagellum assembly. Several genes with known functions related to organelle assembly were found in our samples. *RSPH6A* (radial spoke head 6 homolog A) is a gene that is required for sperm flagellum formation. *RSPH6A* knockout (KO) male mice are infertile as a result of their short immotile spermatozoa [55]. *SPACA1* (sperm acrosome membrane-associated protein 1) gene, is involved in acrosomal morphogenesis and in sperm-egg binding and fusion. *SPACA1* gene-disrupted male mice were infertile and showed abnormal shaping of spermatozoa leading to globozoospermia [56]. Seven genes encoding cilia and flagella associated proteins (CFAP), including *CFAP65* (cilia and flagella associated protein 65), have been related, in humans, with sperm flagellar defetcs and acrosomal hypoplasia [57]. Our dataset contained 14 genes encoding centrosomal proteins, among them we found the *CEP192* (centrosomal protein 192) gene which is involved in protein recruitment during both centrosome duplication and maturation [58].

Epithelium development, which is the progression of an epithelium over time, from its formation to the mature structure, was enriched in the ovine sperm samples with 198 genes. Among them, 3 genes encoding cadherins and cadherin receptors were included, *CELSR1* (cadherin EGF LAG seven-pass G-type receptor 1), *CELSR3* (cadherin EGF LAG seven-pass G-type receptor 3) and *CDHR2* (cadherin related family member 2). Cadherins are expressed in the mature gametes and facilitate the capacitation of sperm in the female reproductive tract and gamete contact during fertilization [59]. Four genes belong to the metallopeptidases family, *ADAM17*, *ADAMTS12*, *ADAMTS16* and *ADAMTSL2*. The metalloprotease *ADAM17* (TACE) has been identified as a putative physiological inducer of germ cell apoptosis [60]. Variants in the human *CCDC103* (coiled-coil domain containing 103) gene have been associated with primary ciliary dyskinesia (PDC), a ciliopathy caused by anomalies in motile cilia which produces subfertility or infertility [61]. Tha *SERPINE2* (serpin peptidase inhibitor, clade E, member 2) gene has broad antiprotease activity specific to serine proteases and is widely expressed in several tissues with the highest levels found in seminal vesicles [62]. Lu and colleagues [62] demonstrated the ability of *SERPINE2* to inhibit sperm capacitation by blocking the cholesterol efflux from sperm plasma membranes and suppressing the increase in the level of sperm protein tyrosine phosphorylation thereby suggesting the role of *SERPINE2* as a sperm decapacition factor [63]. In cattle, SNPs in the *SERPINE2* gene have been associates to the oocyte's cleavage rate [64].

IPA analyses of genes detected in sperm samples highlighted 4 top canonical pathways: Protein Kinase A signaling, Protein Ubiquitination, RAR activation and AMPK signaling. Regarding the Protein Kinase A signaling pathway, 5 genes encoding for adenylyl cyclases (*ADCY1*, *ADCY10*, *ADCY2*, *ADCY6*, *ADCY9*) were highly abundant in the rams' sperm cells. In vertebrates, cAMP is synthesized by two types of adenylyl cyclases, a ubiquitous family of transmembrane adenylyl cyclases with 9 members (*ADCY 1–9*) and a soluble adenylyl cyclase encoded by a single gene *ADCY10* which is alternatively spliced into multiple isoforms [65]. Adcy10 protein was originally thought to be restricted to testis and sperm, but more recently it has been also identified in other cell types [66]. Genetic approaches have demonstrated that *ADCY10* is necessary for male fertility and more specifically, for sperm motility and capacitation [67]. Capacitation is associated with the activation of a cAMP/PKA-dependent signaling pathway leading to phosphorylation of Ser/Thr residues in PKA substrates followed by up-regulation of protein tyrosine phosphorylation [68].

The ubiquitin-proteasome is the major pathway for the selective degradation of abnormal proteins in the cytosol and nucleus. In spermatogenesis, the replacement of histones by protamines is vital for normal sperm formation, which involved the ubiquitination of some enzymes expressed in testis [69]. Recently, histone ubiquitin ligases have been shown to play critical roles in several aspects of spermatogenesis, such as meiotic sex chromosome inactivation, DNA damage response, and spermiogenesis. For this pathway we found molecules such as *UBR1* (Ubiquitin Protein Ligase E3 Component N-Recognin 1), which in mammals is a homologue of *UBR2* gene. In the spermatocytes of ubr2 -/- mice, ubiquitination cannot be induced during meiosis, which seriously affects the chromosome-wide transcriptional silencing of genes linked to unsynapsed axes of the X- and Y-chromosomes. The infertility likely results from *UBR2* histone ubiquitination insufficiency triggering the pachytene checkpoint system [69]. High number of heat shock genes (HSP and DNAJ families) were found in this pathway (S5 Table). Most of these Hsps function as molecular chaperones that prevent the accumulation of aggregated proteins or promote refolding of misfolded proteins. In eukaryotic cells, ubiquitin and certain ubiquitin-conjugating enzymes are Hsps that function in the rapid turnover of denatured proteins [70].

The RAR activation pathway (Retinoic Acid Receptor pathway) is necessary for normal spermatogenesis and epididymal function [71]. Spermatogenesis in mammals is a process partially regulated by testosterone and retinoic acid (RA). RA signaling is mediated by two families of nuclear receptors: retinoic acid receptors (RARs) and retinoid X receptors (RXRs). Four RAR activation pathway related genes (*RARA*, *RARB*, *RXRA* and *RXRB*) were found in our study. *RARA* is expressed at many stages of embryogenesis [72] and its mutations cause sterility in mouse. Mutations in some of the other receptors have yielded male reproductive defects as well. Also, mutation of the gene *RXRB* results in male sterility, due to oligoasthenoteratozoospermia [73].

A recent study has elucidated novel biological actions of the AMPK signaling pathway that are highly relevant for male reproduction [74]. The role of AMPK in cell energy homeostasis makes this kinase crucial in the regulation of spermatozoa functions such as motility, acrosome reaction, and fertilization, which are very dependent on energy levels [75]). AMPK appears to be involved in the mechanisms protecting the cells against the toxic effects of Reactive Oxygen Species (ROS). Four genes directly involved in the AMPK pathway, *TSC1*, *TSC2*, *SIRT1* and *STK11* were found in rams' sperm samples.

### Differentially abundant genes in sperm related to heat stress

In several livestock species, variation on semen quality and reproductive efficiency has been observed linked to the prevailing climatic conditions. Along the hot summer months, the number of sperm cells and their motility decline and the amount of morphological abnormalities increase [76]. This effect has been related to heat stress. The molecular mechanisms underlying this phenomenon remain unclear although may be linked to oxidative stress, DNA damage, apoptosis, autophagy and reduction of mitochondrial activity [77]. Differential expression of genes between warm and thermo-neutral temperatures can shed light on the molecular mechanisms underlying the HS response. However, as mature spermatozoa are assumed to be transcriptionally silent, the RNA load present in these cells, should have been produced in previous stages of the spermatogenic process. Thus, and regarding the HS response, in the mature spermatozoa we are able to assess DA genes which were induced/ repressed by abiotic factors, such as warm events, occurred in the near past and which exert a protection of the spermatozoa from climatic injuries that may occur in the present. In this sense, our group, [5] hypothesized that HS events occurring at initial stages of the

spermatogenesis process (45 to 50 days before sperm collection) would be selectively advantageous to protect subsequent cell types—which transcription and metabolism are more susceptible to HS—against future stress challenges.

In this work, a proportion of the 92 down-regulated annotated genes under HS, are related with reproductive functions, cell growth, cellular cycle arrest, transcription inhibition, lipid metabolism and membrane permeability. A similar trend concerning downregulation of genes as result of thermal response was found in bovine granulosa cells [78]. In this cattle study, the vast majority (1,036) of the 1,211 DA genes found under heat stress, were downregulated. However, in boar sperm, thermal stress seems to mostly promote gene up-regulation [3]. The differences in the number and proportion of down/up regulated genes resulting from HS detected in the different studies in livestock, could be—at least partially—caused by the differences in the climatic conditions during certain stages of spermatogenesis (spermatocytogenesis) rather than by the ambient temperatures at ejaculation [4, 5]. Thus, transcript abundances detected in sperm cells are somehow reflecting warm episodes occurring in the near past, when cells can response to high temperatures inducting/repressing the expression of certain genes.

**Downregulated genes under heat stress conditions.** Some downregulated genes as effect of HS are involved in reproductive functions. The *NOTCH2* (Notch Receptor 2) gene is a component of the Noch signaling pathway. Dirami and colleagues [79] demonstrated that Notch signaling protein components are present in the mouse testis, suggesting an important role for Notch in the regulation of spermatogenesis. The *ALCAM* (Activated Leukocyte Cell Adhesion Molecule) gene is transiently expressed on gonocytes and participates in the migration and cell-to-cell adhesion of gonocytes with Sertoli cells [80]. Its potential involvement in temperature homeostasis has not been reported, but it is a target gene of the transcription regulator *TCF4* which is predicted to be inhibited by high temperatures in the IPA analysis. *SCN10A* (Sodium Voltage-Gated Channel Alpha Subunit 10) encodes the tetrodotoxin-resistant voltage-gated sodium channel alpha subunit, Na v1.8. In humans, *SCN10A* expression is 100-fold higher in testis and sperm than in other tissues so it is considered as testis and sperm-specific Na+ channel with important roles in the regulation of male reproduction [81]. Its predominant localization in the sperm's neck and the principal piece of the flagellum suggests that Na v1.8 could be involved in the modulation of flagellar activity and sperm motility, which is a critical parameter for sperm function and male fertility [82]. The Human Phenotype Ontology for *SCN10A* reports its association with different traits including abnormality of prenatal development or birth (HP: 0001197) and abnormality of temperature regulation (HP: 0004370). Increases in temperature produce a large increment in the energy efficiency through Na+ channel inactivation, which result in a marked decrease in excess Na+ entry [83]. For all this, we can hypothesize that *SCN10A* may be deregulated in sperm cells to cope with cellular effects exerted by high temperatures.

The *TSHZ2* (Teashirt Zinc Finger Homeobox 2) gene encodes a protein predicted to act as a transcriptional repressor. Zhang and colleagues [84] found that this gene was downregulated in samples from testis of rams subjected to a low energy diet. Taken together, these results suggest that the downregulation of *TSHZ2* under HS conditions could be linked to the decrease of feed intake which occurs when animals are subjected to high temperatures. *CYP51A1* (Cytochrome P450 Family 51 Subfamily A Member 1, also named Lanosterol 14-Alpha Demethylase) is a member of the cytochrome P450 superfamily of enzymes which catalyze many reactions involved in drug metabolism and synthesis of cholesterol, steroids and other lipids. In animals, the gene product, P45014DM, catalyzes the lanosterol 14α-demethylase reaction, which is an essential step in cholesterol biosynthesis. *CYP51A1* is highly expressed in the postmeiotic haploid spermatids [85]. Postmeiotic male germ cells contain P45014DM which have

the capacity to synthesize sterols [86] that act as signaling molecules to activate meiosis in spermatogenesis. Inhibition of *CYP51A1* decreases cholesterol levels [87], a fact that may be related with cell cycle arrest observed under HS conditions. In biological cell membranes, cholesterol, along with specific fatty acids, regulates the function of membrane-bound proteins and plays a role in thermal adaptation because of its membrane-stabilizing effect. In Holstein bulls, ejaculates collected in the summer period presented lower cholesterol concentrations than the winter samples [88]. In bovine granulosa cells, downregulation of genes related with steroidogenesis under HS events have been reported [89].

**Upregulated genes under heat stress conditions.** In our study, we found 6 upregulated genes with log2FoldChange ≥1.5. As some of these genes are novel genes in the ovine genome, we considered their orthologues in human, mouse or cow. The upregulated genes list includes *RGS2* (Regulator of G Protein Signaling 2), which is highly expressed in prostate and testis and is involved in spermatogenesis (GO: 0007283). The promoter region of *RGS2* contained a binding sequence for HSF1 (Heat Shock Transcription Factor 1) [90] which is a DNA-binding transcription factor that plays a central role in the transcriptional activation of the heat shock response (HSR). HSF1 promotes the expression of a large class of molecular chaperones heat shock proteins (HSPs) that protect cells from cellular insults. Therefore, the upregulation of *RGS2* gene in sperm cells here observed may be induced by HSF1. *RASIP1* (Ras Interacting Protein 1) was also upregulated in sperm samples collected at high temperatures. *RASIP1* is a critical mediator of endothelial junction stabilization. Endothelial cells constantly remodel their cell–cell junctions and the underlying cytoskeletal network in response to exogenous signals. This remodeling is controlled by a complex signaling network consisting of small G proteins and their various downstream effectors as those encoded by *RAP1* and its effector *RASIP1* genes [91]. High levels of sperm ROS produced by HS can cause sperm dysfunction by the impairment of vascular endothelial function. RAP1 is an important modulator of cell–matrix and cell–cell adhesions in many cell types [92]. Thus, the upregulation of *RASIP1* as response to hyperthermia would induce the activation of RAP1 to contribute to cell junctions' maintenance. The Ensembl gene ENSOARG00000007848 (Musashi RNA Binding Protein 2), was also included in the group of upregulated genes under HS. The Musashi RNA Binding Protein 2 (*MSI2*) encodes an RNA-binding member of the Musashi protein family, which are critical regulators of testis germ cell development and meiosis. It is also essential for sperm development and reproductive potential in mice [93] and it has been involved in stem cell function and cell fate determination in the mammalian system [94]. Sutherland and coworkers demonstrated that the abnormal expression of *MSI2* causes infertility in mice by an overall increase in DNA damage and apoptosis [95]. *MSI2* is expressed during the meiotic and post-meiotic stages of spermatogenesis, consequently MSI2 is functional in later spermatogenesis, playing a key role in the differentiation and development of spermatocytes and spermatids. Thus, overabundance of this gene in rams' ejaculates could be reflecting HS events happened in previous stages of the spermatogenesis process that could be advantageous to cope with future thermal injuries.

Eleven of the 98 DA genes were included in the enriched core of the GSEA analyses: 2 upregulated, *RGS2* and *RASIP1*, and 9 downregulated, *CD46*, *SCHCBP1L*, *BANK1*, *FER*, *RPL17*, *RPS15A*, *ND4L*, *ATP8* and *KPNA4*. *SCHCBP1L* and *CD46* genes were included in several GO terms related with reproduction (reproductive processes, sexual reproduction, male gamete generation and spermatogenesis). In a knock out mouse model to investigate the function of *SHCBP1L* (SHC binding and spindle associated 1 like) gene, Liu and co-workers [96] found that litter size was reduced by loss of Shcbp1l protein and that sperm counts were significantly reduced in homozygotes. These authors also showed that *SHCBP1L* and *HSPA2* (Heat Shock 70kDa Protein 2, also named Epididymis Secretory Sperm Binding Protein) form a complex

that is involved in the maintenance of spindle stability during meiosis. In our study we found a downregulation of the *SHCBP1L* gene (-1.5 Log2Fold Change), but we did not detect any *HSPA2* transcript, even though the presence of RNA from this gene has been demonstrated in human sperm [97]. We don't know what caused the expression decrease of *SHCBP1L* but we cannot rule out a partial relation with the *HSPA2* response to HS. *CD46* (Membrane Cofactor Protein CD46, Trophoblast-Lymphocyte Cross-Reactive Antigen, CD46 molecule) is a ubiquitously expressed complement regulatory protein that protects host cells from complement attack. The protein encoded by this gene may be involved in the fusion of the spermatozoa with the oocyte during fertilization [98]. Variations in *CD46* expression on sperm have been associated with infertility [99]. Rabbit and mouse antibodies against *CD46* inhibit both binding and penetration of human spermatozoa to zona-free hamster eggs and to human zona pellucid [100]. The expression pattern of *CD46* in rodents also supports its role in fertilization. Mice, rats, and guinea pigs express *CD46* solely in spermatozoa [98]. *FER* (FER tyrosine kinase) were present in several GO terms related with the cytoplasmic membrane, protein phosportilation and positive regulation of cellular component organization. *FER* is a member of the Src family of tyrosine kinases regulating numerous cellular processes, including cytoskeletal reorganization, cell adhesion, vesicular transport, and intracellular signaling [101]. *FER* is expressed in testis and has a role in spermatogenesis [102] and it also regulates tyrosine phosphorylation during sperm capacitation [103].

*ND4L* (NADH dehydrogenase subunit 4L), *ATP8* (ATP synthase F0 subunit 8) and *KPNA4* (karyopherin subunit alpha 4) were all downregulated under thermal stress conditions. These genes were enriched in GO terms related with mitochondrial envelope and membrane and ATP metabolic processes. Oxidative stress-induced DNA damage severely affects sperm quality and leads to infertility. Mitochondria are a major source of reactive oxygen species (ROS). In human spermatozoa, physiological levels of ROS play important roles in sperm function, acrosome reaction, capacitation, hyper-activation and the penetration of oocyte by spermatozoa. The energetic activity of mitochondria changes after a heat shock because the respiratory chain is highly thermosensitive. Heat stress inhibits mitochondrial ATP synthesis and results in the dysfunction of the electron transport chain [104]. Spermatozoa require large levels of energy for their survival and proper functioning. Thus, large numbers of mitochondria are uniquely placed in the sperm's midpiece to provide energy quickly and effectively for sperm motility [105]. Specific point mutations and deletions in the mitochondrial DNA (mtDNA) have been associated with poor sperm motility and semen quality in several studies. Mutations in the mitochondrial genes *COXII*, *ATPase6* and *ATPase8* can disrupt ATP production and affect spermatogenesis and sperm motility [106]. *KPNA4* gene, also named Importin Subunit Alpha-4, encodes a protein that plays a central role in the transport of cargo from the cytoplasm to the nucleus. Male knock-out mice for importin α4 were subfertile and yielded smaller litter sizes than wild-type males, indicating that this gene is critical for an appropriate spermatozoa morphology in mice [107]. *BANK1*, *RPL17* and *RPS15A*, which were downregulated under HS, are present in the enriched GO terms related with the regulation of protein, peptides and amide biosynthesis, metabolic processes and translation. *RPL17* and *RPS15A* encode protein components of the 60S and 40S subunits of the ribosomes, respectively. Subjecting mammalian cells to HS inhibits translation of most cellular mRNAs, however few mRNAs, including those related with the HS response, can scape inhibition [108]. This fact involves the decrease in the number of active ribosomes [109] which agrees with the repression of ribosomal genes expression.

IPA analysis of the DA genes showed an enrichment (z_scores $\leq$ -2) of three canonical pathways: Oxidative Phosphorylation, Mitochondrial Dysfunction and PFKFB4 Pathway. The first and second pathways included the *ATP5F1B*, *MT-ATP6*, *MT-ND2* and *MT-ND4L*

mitochondrial genes, and the third pathway involved *NCOA3* and *TGFB3*. All the mitochondrial genes were down-regulated under HS (log2FoldChange between -1 and -1.5) which suggests that HS markedly alters intracellular energetics, characterized by a decrease in ATP production via oxidative phosphorylation and an increase in energy production via aerobic glycolysis [110]. In relation to the PFKFB4 pathway, *PFKFB4* gene encodes the testes 6-phosphofructo-2-kinase which has a strong response to hypoxic stimulation [111]. *PFKFB4* expression is restricted to the spermatogenic cells, being the only isozyme of this family present in mature spermatozoa [112]. In the IPA analysis *PFKFB4* appears as an activated upstream regulator of *NCOA3*, a gene that in this work was downregulated under heat stress (log2FoldChange = -1.96).

Other putative upstream regulators according to IPA included *TCF4*, *SIRT3* and *STAT3*. *TCF4* is a transcription regulator predicted with IPA as inhibited, which target genes were *ALCAM*, *NOTCH2*, *TXNDC11* and *URI1*, all of them found downregulated in sperm samples collected under HS conditions. *TCF4* is connected to the FOXO1 pathway which regulates whole-body energy balance and plays a role in oxidative stress, protein turnover, immunity modulation, glucose homeostasis, mitochondrial function, AKT signaling, mTOR signaling, WNT signaling, and insulin signaling [113]. The *SIRT3* (Sirtuin 3 also named NAD-Dependent Protein Deacetylase Sirtuin-3) gene encodes a member of the sirtuin family of class III histone deacetylases. *SIRT3* is localized exclusively in mitochondria and is considered to be a key regulator of many cellular functions, including stress resistance, energy metabolism, apoptosis and aging. *SIRT3* activates mitochondrial functions and plays an important role in adaptive thermogenesis by increasing mitochondrial respiration [114]. Caloric restriction activates *SIRT3* expression in both white and brown adipose. Additionally, cold exposure up-regulates SIRT3 expression in brown fat, whereas elevated climate temperature reduces its expression [114]. Their dependence on NAD+ links the activity of this gene to the metabolic status of the cell, maintaining intracellular levels of NAD+ which is crucial for the management of stress response of cell, by inducing the expression of antioxidants to reduce cellular ROS levels [115]. The IPA analysis predicted an activation of this upstream regulator with a z_score close to the significance threshold (1.987). However, in our study, we detected a suggestive but non-significant downregulation of SIRT3 under HS conditions (log2FoldChange = - 0.6). However, all the IPA target genes of this enzyme, *MT-ATP6*, *MT-ATP8*, *MT-ND2* and *MT-ND4L*, resulted significantly downregulated under HS. *STAT3* (Signal Transducer and Activator of Transcription 3) is a transcriptional regulator that mediates the expression of a variety of genes in response to cell stimuli, and thus plays a key role in many cellular processes such as cell growth and apoptosis. *STAT3* has been identified in human [116] and porcine [117] spermatozoa. However, the presence of STAT proteins in sperm structural components suggests that their role is different from their well-known transcription factor activity in somatic cells [116]. *STAT3* (also named Acute Phase Response Factor) is crucial for maintaining temperature homeostasis [118] and becomes activated by various cytokines and regulates the expression of genes essential for systemic thermoregulation, embryogenesis, and immune response [119]. Lachance and Leclerc [120], showed that six of the seven known STAT proteins, namely *STAT1*, *3*, *4*, *5A*, *5B*, and *6*, are present in the sperm flagellum, and that *STAT3*, although was detected in the fraction containing mostly sperm head plasma membranes, was strongly enriched in the flagellum. Weyrich and coworkers [121] verified that males exposed to an increased ambient temperature for 2 months produced DNA methylation and changes in the expression of *STAT3* in wild guinea pigs offspring. STAT3 was hyper methylated as result of high temperature treatment and also showed a significant expression reduction in the progeny.

## Conclusion

In this work, 80 ejaculates from 40 rams of the Spanish Manchega breed, collected in two seasons with differential climatic conditions (heat and comfort) were used to assess the core sperm transcriptome and its changes due to seasonal factors. In medium-latitude regions, as is the case of the region in the Iberian Peninsula described in our study, the fertility of rams shows less seasonal breeding behavior affected by the photoperiod [122]. Therefore, climatic factors must be the most important determinants of the changes observed in the sperm transcriptome. In this work we have found that despite the transcriptional inactivity of sperm cells, rams spermatozoa carry thousands of RNAs which are not merely residual products from the spermatogenesis process, but have critical functions for the spermatozoa functionality and surviving and for egg fertilization and embryo development. Sperm are one or the most sensitive cells to heat stress effects, being its response to this stressor a drastic decrease of the transcriptional activity, and the upregulation of a few genes related with the basic functions to maintain the organisms' homeostasis and surviving. We can conclude that rams' spermatozoids carry remnant mRNAs which are retrospectively indicative of events occurring along the spermatogenesis process, including abiotic factors (environmental temperature), but also, transcripts that might play vital roles in fertilization and embryogenesis.

## Supporting information

**S1 Fig. ZIMB-WaVE model with and without run as sample-level covariate.** a) with climatic condition (control (O) and heat stress (J) as sample-level covariate and b) with climatic condition and run as sample-level covariates. In both cases K = 2, épsilon = 104, V = 1.
(TIF)

**S1 Table. Sequencing summary.**
(XLSX)

**S2 Table. Genes and transcripts detected in rams' sperm samples.**
(XLSX)

**S3 Table. GSEA results for all genes/transcripts detected in rams' sperm samples.**
(XLSX)

**S4 Table. Significant GO terms for all genes/transcripts detected in rams' sperm samples.**
(XLSX)

**S5 Table. IPA analysis of all genes/transcripts detected in rams' sperm samples.**
(XLSX)

**S6 Table. DA genes between sperm samples collected in variable climatic conditions and GSEA analysis.**
(XLSX)

**S7 Table. IPA analysis of DA genes between samples collected in variable climatic conditions.**
(XLSX)

## Acknowledgments

We are grateful to the Breeders Association of Manchega sheep breed (AGRAMA) and to the Regional Centre of Animal Selection and Reproduction from Valdepeñas (CERSYRA) for providing animals' samples and laboratory facilities to process ejaculates, and to the Super computation Centre from Galicia (CESGA) to provide computation facilities.

## Author Contributions

**Conceptualization:** Magdalena Serrano.

**Data curation:** Irene Ureña, Carmen González.

**Formal analysis:** Irene Ureña, Manuel Ramón, Marta Gòdia, Alex Clop.

**Funding acquisition:** Magdalena Serrano.

**Investigation:** Irene Ureña, Manuel Ramón, Marta Gòdia, Alex Clop, Magdalena Serrano.

**Methodology:** Irene Ureña, Carmen González, Manuel Ramón, Marta Gòdia, Alex Clop, Jorge H. Calvo, Magdalena Serrano.

**Project administration:** Magdalena Serrano.

**Software:** Irene Ureña, Manuel Ramón, Marta Gòdia, Alex Clop.

**Supervision:** Jorge H. Calvo.

**Visualization:** Mª Jesús Carabaño.

**Writing – original draft:** Irene Ureña, Marta Gòdia, Alex Clop, Jorge H. Calvo, Magdalena Serrano.

**Writing – review & editing:** Carmen González, Manuel Ramón, Alex Clop, Jorge H. Calvo, Mª Jesús Carabaño, Magdalena Serrano.

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
