## [Decision Letter · Decision Letter 0]

6 Jan 2022

PONE-D-21-37178Exploring the ovine sperm transcriptome by RNAseq techniques. I Effect of climatic conditions on transcripts abundancePLOS ONE

Dear Dr. Serrano,

Thank you for submitting your manuscript to PLOS ONE. After careful consideration, we feel that it has merit but does not fully meet PLOS ONE’s publication criteria as it currently stands. Therefore, we invite you to submit a revised version of the manuscript that addresses the points raised during the review process.

Please submit your revised manuscript at your earliest convenience. Please include the following items when submitting your revised manuscript:A rebuttal letter that responds to each point raised by the academic editor and reviewer(s). You should upload this letter as a separate file labeled 'Response to Reviewers'.A marked-up copy of your manuscript that highlights changes made to the original version. You should upload this as a separate file labeled 'Revised Manuscript with Track Changes'.An unmarked version of your revised paper without tracked changes. You should upload this as a separate file labeled 'Manuscript'.

We look forward to receiving your revised manuscript.

Kind regards,

Joël R Drevet, Ph.D.

Academic Editor

PLOS ONE

Journal Requirements:

"RTA2013-00041 INIA project has provided the funding to do this work."

"The authors received no specific funding for this work."

Additional Editor Comments:

Two reviewers found some value in this submission but requested several improvements that, if considered, would strengthen this report. Please consider each of these when reviewing your manuscript.

Reviewers' comments:

Reviewer's Responses to Questions

**Comments to the Author**

1. Is the manuscript technically sound, and do the data support the conclusions?

Reviewer #1: Partly

Reviewer #2: Yes

2. Has the statistical analysis been performed appropriately and rigorously? 

Reviewer #1: Yes

Reviewer #2: Yes

3. Have the authors made all data underlying the findings in their manuscript fully available?

Reviewer #1: No

Reviewer #2: Yes

4. Is the manuscript presented in an intelligible fashion and written in standard English?

Reviewer #1: No

Reviewer #2: Yes

5. Review Comments to the Author

Reviewer #1: In this study, Urena et al., reported RNA-seq results of ovine sperm transcriptomes from 64 ejaculates. Among them 28 were collected in summer and 36 were collected in autumn, allowing them to evaluate the seasonal effects on the sperm transcriptome. The authors performed functional enrichment analyses on total sperm transcripts, as well as on differentially abundant (DA) transcripts, and identified several enriched functional categories. Once published, these sequencing data will provide the community with very useful resources of ovine sperm RNAs. However, before that, I have some questions and comments listed below.

1. I didn’t find ‘figure legends’ in the manuscript, which made it hard for me to understand each figure.

2. Figure 1 looks confusing. The numbers from the pie chart didn’t add up to the number(s) of protein coding genes / transcripts shown at the bottom-right corner. Did the authors happen to exclude the largest category, the protein-coding mRNAs?

3. For Figures 2 and 5, if GSEA results are shown here, it would be better to show NES (Enrichment score) and p value, rather than the bar plots of percentages. Also, without figure legend, I don’t understand what the percentages mean.

4. The raw sequencing data need to be made publicly available upon publication of the study. The authors can upload the data to databases such as ‘Gene Expression Omnibus (GEO)’, and add a paragraph of ‘Data Availability Statement’ to the manuscript.

5. The authors only identified 98 differentially abundant genes, by using the criteria of Padj < 0.05 and Log2FC >= 1.5. (I am not sure if I understand correctly, as in L263 the authors wrote >= 1.5, but in L268 they wrote >= 2.0. Please double check)

Here, the cutoff for fold changes is pretty harsh (2^1.5 = 2.83 fold). I would suggest use a milder cutoff, for example Log2FC >=1 so that’s 2 fold. In this case the authors may get more DA genes and make more sense of the functional differences. But this is optional.

Reviewer #2: In this submission, a study of the sperm ovine transcriptome has been developed in the view to get knowledge of the different sperm RNAs abundances and their potential modifications during heat variation between summer and autumn.

An important sequencing analysis of ram sperm RNAs of 40 animals is presented with differences according to the temperature of the environment surrounding of the animals. More than 10 000 transcripts have been obtained and 236 genes were differently expressed between summer and autumn..

General comments:

Important results about the sperm RNAs are presented in this submission. The analysis of the results has been done with probably the best methods but a lot of information is need.

First of them is relative to the characteristic of the sperm collected for this analysis before purification. Nowhere information is given such as concentration, morphology, mobility, presence or not of the cytoplasmic droplets etc. for the samples obtained in July and October.

Furthermore as these samples came from an artificial insemination centers, fertility index of the animals during summer and autumn should be probably presented between these two seasons.

The choice of the two seasons is based on the temperature but what about the variation of the endocrinology of the animals induced with the photo period?

The sequencing methodology is unclear. The RNAs from 24 animals have been sequenced but how the analysis has been done? Each couple of sperm samples has been analysed by animal or a mix has been done for each season? What is the variation of the results between animals? How much RNAs are common between animals?

Information of the length of the sequence should be given somewhere.

In the discussion, the variations of the RNAs should be discussed with the parameters of the sperm characteristics.

Ref 26 is not associated with translational activity IN the sperm cells.

6. PLOS authors have the option to publish the peer review history of their article (what does this mean?). If published, this will include your full peer review and any attached files.

Reviewer #1: No

Reviewer #2: No

---

## [Author Response · Author response to Decision Letter 0]

17 Jan 2022

Response to Reviewers

Journal Requirements:

Response: We have upgraded the manuscript formatting based on the requirements (e.g. Figure has been replaced by Fig; Supplementary Legends tittles are now in bold).

"RTA2013-00041 INIA project has provided the funding to do this work."

Response: Ok. Sorry for the mistake. Funding-related text has been removed from the manuscript. For this work we declare that "The authors received no specific funding for this work", as figure in the online submission. In the ACNOWLEDGEMENTS section we have add the following sentence: “This work has been supported by the RTA2013-00041 INIA project”.

Reviewer #1: In this study, Urena et al., reported RNA-seq results of ovine sperm transcriptomes from 64 ejaculates. Among them 28 were collected in summer and 36 were collected in autumn, allowing them to evaluate the seasonal effects on the sperm transcriptome. The authors performed functional enrichment analyses on total sperm transcripts, as well as on differentially abundant (DA) transcripts, and identified several enriched functional categories. Once published, these sequencing data will provide the community with very useful resources of ovine sperm RNAs. However, before that, I have some questions and comments listed below.

1. I didn’t find ‘figure legends’ in the manuscript, which made it hard for me to understand each figure.

We apologize, there must have been some error during the uploading of the manuscript. Figure legends have been added to the main manuscript after the paragraph in which they are first cited.

2. Figure 1 looks confusing. The numbers from the pie chart didn’t add up to the number(s) of protein coding genes / transcripts shown at the bottom-right corner. Did the authors happen to exclude the largest category, the protein-coding mRNAs?

Thanks for pointing out this mistake. Percentages for the different kind of genes/transcripts in Figure 1 were indeed wrong. They were calculated without taking into account the total genes detected. An updated Figure 1 has been created. Protein coding genes account for the 91% of the total genes/transcripts detected, and the remaining (approx. 9%), corresponded to miRNA, miscRNA, Mt rRNA, snoRNA, etc. We have included the acronyms meaning in the figure legend.

3. For Figures 2 and 5, if GSEA results are shown here, it would be better to show NES (Enrichment score) and p value, rather than the bar plots of percentages. Also, without figure legend, I don’t understand what the percentages mean.

We have updated Figures 2 and 5 as the reviewer suggested. In the new figures, additional tables with GO terms IDs and names, number of genes in each GO term, NES values, nominal p_values and FDR values have been included.

4. The raw sequencing data need to be made publicly available upon publication of the study. The authors can upload the data to databases such as ‘Gene Expression Omnibus (GEO)’, and add a paragraph of ‘Data Availability Statement’ to the manuscript.

Raw data of this work has been submitted to the NCBI repository with the Bio Project accession number PRJNA733107. We have included a “Data availability” statement after the supplementary material legends:

L1099-1100: “Data availability: The datasets generated and analysed in the current study are available at NCBI’s BioProject PRJNA733107.

5. The authors only identified 98 differentially abundant genes, by using the criteria of Padj < 0.05 and Log2FC >= 1.5. (I am not sure if I understand correctly, as in L263 the authors wrote >= 1.5, but in L268 they wrote >= 2.0. Please double check).

Here, the cutoff for fold changes is pretty harsh (2^1.5 = 2.83 fold). I would suggest use a milder cutoff, for example Log2FC >=1 so that’s 2 fold. In this case the authors may get more DA genes and make more sense of the functional differences. But this is optional.

Thank you for your comments. In the case of sperm transcriptome and since the difficulties to detect differences in gene expression we have decided to choose a high logFC threshold to avoid having false positive DA genes. We preferred to have less DA genes, but real, than higher DA genes that could include several false positives that could hamper the real biological effects.

Sorry for the mistake. Effectively, we have considered the threshold log2FC > 1.5 to declare a gene as DA. However, in Figure 4 we only showed those DA genes with a log2FC > 2, but only for a practical question, and that is that they could be represented graphically, otherwise the Figure was too big. We are making this clear in the Figure 4 legend:

Fig 4. Upregulated (dark grey) and down-regulated (light grey) protein coding genes under heat stress conditions. The DA genes passed the filters of padj < 0.05 and absolute log2FoldChange ≥ 1.5. In this figure, we only show genes that presented padj < 0.05 and absolute log2FoldChange ≥ 2.” 

Reviewer #2: In this submission, a study of the sperm ovine transcriptome has been developed in the view to get knowledge of the different sperm RNAs abundances and their potential modifications during heat variation between summer and autumn.

An important sequencing analysis of ram sperm RNAs of 40 animals is presented with differences according to the temperature of the environment surrounding of the animals. More than 10 000 transcripts have been obtained and 236 genes were differently expressed between summer and autumn

General comments: Important results about the sperm RNAs are presented in this submission. The analysis of the results has been done with probably the best methods but a lot of information is need.

First of them is relative to the characteristic of the sperm collected for this analysis before purification. Nowhere information is given such as concentration, morphology, mobility, presence or not of the cytoplasmic droplets etc. for the samples obtained in July and October.

Sperm characteristics of the ejaculates collected in these rams for this work, are those routinely measured in the artificial insemination centre. These sperm quality traits were ejaculate volume, the spermatozoa concentration and mass motility (evaluated on a 0 to 5 scale). No data about morphology and cytoplasmic droplets have been collected. All ejaculates used in this study met the quality requirements set at the centre for use in AI.

Furthermore, as these samples came from an artificial insemination centres, fertility index of the animals during summer and autumn should be probably presented between these two seasons.

Effectively data of the rams came from an artificial insemination centre. The rams used in this work are those used for inseminate ewes in all the herds participating in the genetic breeding program of this breed to improve milk traits. However, since to perform this work we used the complete ejaculates of this rams in each season, no data about the fertility index of these ejaculates are available, since no inseminations were performed with them. However, if the reviewer feels it is important to indicate the raw fertility values on each season, we could request such information from the breeders association and the AI centre.

The choice of the two seasons is based on the temperature but what about the variation of the endocrinology of the animals induced with the photo period?

Regarding the endocrinology changes induced as effect of the photoperiod, no measures have been made in this work. It is important to remark that the biological material used in this work came from commercial rams which belonged to the artificial insemination centre and only one ejaculate was provided. Therefore, it is not possible to take and analyse different measures from those routinely collected in the centre.

The sequencing methodology is unclear. The RNAs from 24 animals have been sequenced but how the analysis has been done? Each couple of sperm samples has been analysed by animal or a mix has been done for each season? What is the variation of the results between animals? How much RNAs are common between animals?

The analysis to test differential RNA abundance in July and October samples consisted of a comparison between individual samples taken in each season but including the animal effect to take into account that most samples were taken in the same experimental unit (the animal) and therefore a common environment effect must be considered. . Including this information in the model design will account for differences between the samples while estimating the effect due to the heat conditions. To make it more clear, we now include in the material and methods of this analysis:

Page 6, lines 207-209: “The effect of the animal was also included in the DESEq2 model using a multi-factor design which includes sample information.”

Information of the length of the sequence should be given somewhere.

Characteristics relative to RNA sequencing are fully detailed in S1 Table. HighSeq2000 Illumina platform yields 75bp long paired-end reads. In S1 Table and for each RNA sequence run (OVISPERM_01, 03 and 04) the column "Avg alignment insert size" shows the average sequence length for each sample.

In the discussion, the variations of the RNAs should be discussed with the parameters of the sperm characteristics.

We understand the interest of the reviewer in the analysis of the association between transcripts abundance and sperm characteristics. But, as the title of the work reflects – “Exploring the ovine sperm transcriptome by RNAseq techniques. I Effect of climatic conditions on transcripts abundance”- this is the first manuscript in a series of three regarding a central topic that is the study of the changes in the ram’s sperm transcriptome and metagenome due to climatic effects and their impact in sperm characteristics linked to their reproductive efficiency.

In this large experiment we have collected not only sperm samples from rams in two different seasons but a few sperm quality traits, volume and concentration, and assessed sperm DNA fragmentation assessed by flow cytometry for all the samples collected. The aim of the second part of this work is to analyse the relationship between transcripts abundance and sperm characteristics and also, a variant calling study with the aim to detect variants affecting transcripts abundance in both seasonal conditions. As the part I of this work is already very long, we considered necessary to split the work into three distinct parts. The first that currently concerns us, a deep and discussed analysis of the ram’s transcriptome and the effect of climatic conditions on transcripts differential abundance. For example, this type of work alone has already been published in other species, as in boars, “A RNASeq Analysis to Describe the Boar Sperm Transcriptome and Its Seasonal Changes” Gòdia et al., 2018, and it has helped on later studies related to sperm quality parameters, to discern between real differently abundant genes than from those false positives, identified by differences of heat stress. In the second study, we aim to perform an association analysis between transcripts abundance and sperm traits, and perform a variant calling analysis. And the third, we aim to study of the microbiome transcripts existing in sperm samples and its changes due to seasonal changes.

This is the reason for not including this data, because we want to prepare a second manuscript with the association of transcripts abundance and sperm characteristics as the reviewer very well suggests it. But we prefer to split it, to ensure that the message of the role of heat stress in the ovine transcriptome is clear. We have identified several transcripts altered because of heat stress, and we feel is important for the scientific community to share this results for potential future work on this field.

Ref 26 is not associated with translational activity IN the sperm cells.

Sorry for the mistake. This reference is wrong. We have included the correct reference in the References section. 

26 Gur Y, Breitbart H. Mammalian sperm translate nuclear-encoded proteins by mitochondrial-type ribosomes. Genes Dev. 2006; 20(4):411-416. doi:10.1101/gad.367606

---

## [Decision Letter · Decision Letter 1]

24 Jan 2022

PONE-D-21-37178R1Exploring the ovine sperm transcriptome by RNAseq techniques. I Effect of climatic conditions on transcripts abundancePLOS ONE

Dear Dr. Serrano,

Thank you for submitting your manuscript to PLOS ONE. After careful consideration, we feel that it has merit but does not fully meet PLOS ONE’s publication criteria as it currently stands. Therefore, we invite you to submit a revised version of the manuscript that addresses the points raised during the review process.

Please submit your revised manuscript at Mar 10 2022 11:59PM. If you will need more time to complete your revisions, please reply to this message or contact the journal office at plosone@plos.org. Please include the following items when submitting your revised manuscript:A rebuttal letter that responds to each point raised by the academic editor and reviewer(s). You should upload this letter as a separate file labeled 'Response to Reviewers'.A marked-up copy of your manuscript that highlights changes made to the original version. You should upload this as a separate file labeled 'Revised Manuscript with Track Changes'.An unmarked version of your revised paper without tracked changes. You should upload this as a separate file labeled 'Manuscript'.

We look forward to receiving your revised manuscript.

Kind regards,

Joël R Drevet, Ph.D.

Academic Editor

PLOS ONE

Journal Requirements:

Additional Editor Comments:

One reviewer is only partially satisfied with the proposed revised version. The reviewer raises a major concern that should be appropriately addressed. Currently, this is not the case. Please take this into account and modify your report accordingly. This will be the last time you have the opportunity to respond to the reviewer's comment.

Reviewers' comments:

Reviewer's Responses to Questions

**Comments to the Author**

1. If the authors have adequately addressed your comments raised in a previous round of review and you feel that this manuscript is now acceptable for publication, you may indicate that here to bypass the “Comments to the Author” section, enter your conflict of interest statement in the “Confidential to Editor” section, and submit your "Accept" recommendation.

Reviewer #1: All comments have been addressed

Reviewer #2: All comments have been addressed

2. Is the manuscript technically sound, and do the data support the conclusions?

Reviewer #1: Yes

Reviewer #2: Yes

3. Has the statistical analysis been performed appropriately and rigorously? 

Reviewer #1: Yes

Reviewer #2: Yes

4. Have the authors made all data underlying the findings in their manuscript fully available?

Reviewer #1: Yes

Reviewer #2: Yes

5. Is the manuscript presented in an intelligible fashion and written in standard English?

Reviewer #1: Yes

Reviewer #2: Yes

6. Review Comments to the Author

Reviewer #1: (No Response)

Reviewer #2: According to my comments about the lack of information on the spermatozoa parameters of the samples used for this transcriptomic study, the authors written that these parameters are those of artificial centre. However, according to the authors, these parameters should be given in secund submission. How could a reviewer and a reader interpret such episode publications? It should be better that these publications should be submitted in same time.

Furthermore, all the discussion of the present submission is based on a heat effect but no where the condition of the ram breeding was given. Are the animals in the barn or in open air? Is the temperature given in the text the same in the barn etc.. Numerous publications shown that the temperature regulation of the testis of the ram could be very efficient even in hot environment.

A temperature effect is probable but could not presented in this submission has the only effect on the transcriptome result. The sexual activity of the ram is completely different between the summer and autumn (androgen, sperm production and quality etc..).

I suggest also that the title should be:” Exploring the ovine sperm transcriptome by RNAseq techniques. I Effect of seasonal conditions on transcripts abundance”.

7. PLOS authors have the option to publish the peer review history of their article (what does this mean?). If published, this will include your full peer review and any attached files.

Reviewer #1: No

Reviewer #2: No

---

## [Author Response · Author response to Decision Letter 1]

7 Feb 2022

Response to Reviewers

Additional Editor Comments:

One reviewer is only partially satisfied with the proposed revised version. The reviewer raises a major concern that should be appropriately addressed. Currently, this is not the case. Please take this into account and modify your report accordingly. This will be the last time you have the opportunity to respond to the reviewer's comment.

We would like to thank you for the time and effort in reviewing this work, and for the suggestions made by the reviewers. We also thank the editor for the opportunity to clarify the doubts raised by reviewer 2. We hope that both, the reviewer and the editor, agree with our responses, and if there are no other major concerns, the article will be accepted for publication. 

Reviewer #2

According to my comments about the lack of information on the spermatozoa parameters of the samples used for this transcriptomic study, the authors written that these parameters are those of artificial centre. However, according to the authors, these parameters should be given in second submission. How could a reviewer and a reader interpret such episode publications? It should be better that these publications should be submitted in same time.

We agree with the reviewer that in addition to the characterization of the sheep sperm transcriptome and its seasonal variation, the link of these RNA changes with sperm quality traits is of high interest due to their relevance for the sheep production sector. To help understanding the relevance of our results, we have modified the article accordingly and have now added a table (Table 2) showing the average and standard deviation values for the different semen parameters in each group and also the results of a T-test comparing these values in both groups. The results show that the ejaculate concentration was significantly different in heat vs comfort samples, but not the rest of the traits. This may mean that transcriptional changes between seasons are efficient in protecting sperm cells from possible damage caused by environmental changes, so that sperm characteristics are little affected by them Therefore, the identification of the genes (and the metabolic routes in which they are involved) that show significant changes in their RNA abundance in sperm to cope with seasonal changes is an important step to disentangle its potential effect on sperm characteristics, and by extension, on the reproductive efficiency of the males.

We believe that evaluating the seasonal differences of the ovine sperm transcriptome is of relevance, because it can highlight the crucial genes that cope during changes in environmental conditions during spermatogenesis to grant the efficient production of viable and functional spermatozoa. We are tackling this question from two different angles. One angle, addressed in this article, is to understand the molecular changes that may help coping with seasonal environmental variation. The other angle is to directly relate the abundance of each gene in sperm, with the semen quality in relation to these seasonal changes. We are currently drafting a second manuscript tackling this other angle. 

Furthermore, all the discussion of the present submission is based on a heat effect but nowhere the condition of the ram breeding was given. Are the animals in the barn or in open air? Is the temperature given in the text the same in the barn etc.?

The rams evaluated in this study were kept in standardized conditions in the artificial insemination centres. All the animals were in conditioned boxes within the barn, with natural ventilation and with similar feeding and handling. The animals are allowed to graze outdoors daily, but avoiding the hottest hours during the summer months. The station itself does not have a weather station. Thus, the meteorological conditions that we reported were provided by the closest official meteorological station which shares very similar weather conditions for the short distance and similar altitude. Importantly, this data indicates clear differences in temperature and relative humidity between the hot and the comfort seasons. A paragraph providing this information has been added to the manuscript.

Page 2, lines 95-98: “). Rams were handled in conditioned boxes inside the barn, with natural ventilation and similar feeding and management. The animals are allowed to go outside the stables daily, but avoiding the hottest hours of the summer months. The temperature recorded (Table 1) reflects the outdoor conditions, in order to display the differences between the hot and comfort seasons”.

Numerous publications shown that the temperature regulation of the testis of the ram could be very efficient even in hot environment. A temperature effect is probable but could not presented in this submission has the only effect on the transcriptome result. The sexual activity of the ram is completely different between the summer and autumn (androgen, sperm production and quality etc.).

We fully agree with this statement, and even more so for this breed of sheep which is very well adapted to the climatic conditions of this particular geographical region. This efficient regulation at the physiological and anatomical levels, arises from the fact that in most mammals the testicle, which is contained in the scrotum, is exposed to a temperature that is approximately 3°C lower than the inside body temperature (Hansen PJ. Philos Trans R Soc Lond B Biol Sci. 2009; 364(1534): 3341-50) to protect the sperm cells from heat injuries. However, despite the existence of this efficient temperature regulation in the mammalian testicle, the high sensitivity of sperm cells to environmental heat has been extensively accepted and studied (Perez-Crespo M et al. Mol Reprod Dev. 2008; 75(1):40-7; Grazer VM, Martin OY. Biology (Basel). 2012; 1(2):411-38; Takahashi M., Reprod Med Biol (2012) 11:37–47; Durairajanayagam D. Reproductive BioMedicine Online Vol. 2014, Vol 30(1) p14-27; Shahat A.M. et al. Theriogenology 158 (2020), 84-96). Heat stress arises when the effective temperature of the environment exceeds the animals’ upper critical temperature, which in sheep ranges between 25 and 31 °C, depending on breed, age and physiological state (Hopkins PS et al. Aust J Agric Res. 1978; 29:161–71). The physiological stress caused by hot weather can alter the sperm’s transcriptome (Jannatifar R. et al. Association of heat shock protein A2 expression and sperm quality after N-acetyl-cysteine supplementation in astheno-terato-zoospermic infertile men. Andrologia. 2021 Jun; 53(5):e14024; Lymbery RA. Et al. Post-ejaculation thermal stress causes changes to the RNA profile of sperm in an external fertilizer. Proc Biol Sci. 2020 Nov 11; 287(1938):20202147; Rizzoto G. et al. Acute mild heat stress alters gene expression in testes and reduces sperm quality in mice. Theriogenology. 2020 Dec; 158:375-381) and even the semen quality and fertility of the animals (Jeremy M. et al. Co-treatment of testosterone and estrogen mitigates heat-induced testicular dysfunctions in a rat model J Steroid Biochem Mol Biol. 2021 Nov; 214: 106011; Flowers WL. Factors affecting the production of quality ejaculates from boars. Anim Reprod Sci. 2021 Sep 6:106840; Pirani M. et al. Protective Effects of Fisetin in the Mice Induced by Long-Term Scrotal Hyperthermia. Reprod Sci. 2021 Nov; 28(11):3123-3136; Llamas-Luceño N. et al. High temperature-humidity index compromises sperm quality and fertility of Holstein bulls in temperate climates. J Dairy Sci. 2020 Oct; 103(10):9502-951). The scientific literature suggests that to cope with environmental stressors there is a regulation of the expression of certain genes involved in the heat-stress response (Godìa M, et al. Front Genet. 2019; 10:299; Ramon M, et al. PLoS One. 2014;9(1):e86107. Salces-Ortiz J et al. PLoS One. 2015; 10(2):e0116360. Hansen PJ. Philos Trans R Soc Lond B Biol Sci. 2009; 364(1534):3341-50). 

It is true, as reviewer 2 pointed out, that there are other factors related with the sexual activity of rams along seasons. For example, the photoperiod has an important impact in this sexual activity as it is related to the seasonal anoestrus. However, in medium-latitude regions, commonly found in regions with mild winters, as is the case of the region in the Iberian Peninsula described in our study, the fertility of rams shows less seasonal breeding behaviour affected by the photoperiod; thus, testicular size, sperm production, and mating capacity are mildly reduced during the increased photoperiods (Avdi et al. 2004, Theriogenology, 62, 275–282). Our data regarding sperm quality traits (Table 2) shows that that there were no differences between them due to seasonal changes, except for sperm concentration. Therefore, the potential effect of the photoperiod on the reproductive efficiency of the Manchega rams in our geographical location, and reproductive systems with artificial insemination, is likely to be small. Likewise, the effect of the photoperiod in the transcriptome changes that we observed in our study should be low. 

Regarding this topic we have added in the conclusions of the manuscript the following sentence:

Pages 18-19, lines 724-732. “In this work, 80 ejaculates from 40 rams of the Spanish Manchega breed, collected in two seasons with differential climatic conditions (heat and comfort) were used to assess the core sperm transcriptome and its changes due to seasonal factors. In medium-latitude regions, as is the case of the region in the Iberian Peninsula described in our study, the fertility of rams shows less seasonal breeding behavior affected by the photoperiod [122]. Therefore, climatic factors must be the most important determinants of the changes observed in the sperm transcriptome. In this work we have found that despite the transcriptional inactivity of sperm cells, rams spermatozoa carry thousands of RNAs which are not merely residual products from the spermatogenesis process, but have critical functions for the spermatozoa functionality and surviving and for egg fertilization and embryo development”.

I suggest also that the title should be:” Exploring the ovine sperm transcriptome by RNAseq techniques. I Effect of seasonal conditions on transcripts abundance”.

We fully agree with the opinion of reviewer 2 regarding the title of the manuscript. We have changed the title of the manuscript to the one rightly suggested by reviewer 2.

New title of the manuscript:” Exploring the ovine sperm transcriptome by RNAseq techniques. I Effect of seasonal conditions on transcripts abundance”.

---

## [Decision Letter · Decision Letter 2]

22 Feb 2022

Exploring the ovine sperm transcriptome by RNAseq techniques. I Effect of seasonal conditions on transcripts abundance

PONE-D-21-37178R2

Dear Dr. M. Serrano,

We’re pleased to inform you that your manuscript has been judged scientifically suitable for publication and will be formally accepted for publication once it meets all outstanding technical requirements.

Kind regards,

Joël R Drevet, Ph.D.

Academic Editor

PLOS ONE

Additional Editor Comments (optional):

Reviewers' comments:

Reviewer's Responses to Questions

**Comments to the Author**

1. If the authors have adequately addressed your comments raised in a previous round of review and you feel that this manuscript is now acceptable for publication, you may indicate that here to bypass the “Comments to the Author” section, enter your conflict of interest statement in the “Confidential to Editor” section, and submit your "Accept" recommendation.

Reviewer #2: (No Response)

2. Is the manuscript technically sound, and do the data support the conclusions?

Reviewer #2: Yes

3. Has the statistical analysis been performed appropriately and rigorously? 

Reviewer #2: Yes

4. Have the authors made all data underlying the findings in their manuscript fully available?

Reviewer #2: Yes

5. Is the manuscript presented in an intelligible fashion and written in standard English?

Reviewer #2: Yes

6. Review Comments to the Author

Reviewer #2: This revised presentation has included minimal information’s about the sperm sampling. Seasonal variations have been evocated in the title and the conclusion of this submission.

7. PLOS authors have the option to publish the peer review history of their article (what does this mean?). If published, this will include your full peer review and any attached files.

Reviewer #2: No

---

## [Editor Report · Acceptance letter]

28 Feb 2022

PONE-D-21-37178R2 

Exploring the ovine sperm transcriptome by RNAseq techniques. I Effect of seasonal conditions on transcripts abundance 

Dear Dr. Serrano:

I'm pleased to inform you that your manuscript has been deemed suitable for publication in PLOS ONE. Congratulations! Your manuscript is now with our production department. 

Kind regards, 

on behalf of

Prof. Joël R Drevet 

Academic Editor

PLOS ONE